# Cyanobacteria and Cyanotoxins in a Changing Environment: Concepts, Controversies, Challenges

**Ingrid Chorus [1,*], Jutta Fastner [1] and Martin Welker [2]**

1   German Environment Agency, Schichauweg 58, 12307 Berlin, Germany; jutta.fastner@uba.de
2   Independent Researcher, 10117 Berlin, Germany; martin.welker@online.de
*   Correspondence: ingrid.chorus@gmail.com

**Abstract:** Concern is widely being published that the occurrence of toxic cyanobacteria is increasing in consequence of climate change and eutrophication, substantially threatening human health. Here, we review evidence and pertinent publications to explore in which types of waterbodies climate change is likely to exacerbate cyanobacterial blooms; whether controlling blooms and toxin concentrations requires a balanced approach of reducing not only the concentrations of phosphorus (P) but also those of nitrogen (N); how trophic and climatic changes affect health risks caused by toxic cyanobacteria. We propose the following for further discussion: (i) Climate change is likely to promote blooms in some waterbodies—not in those with low concentrations of P or N stringently limiting biomass, and more so in shallow than in stratified waterbodies. Particularly in the latter, it can work both ways—rendering conditions for cyanobacterial proliferation more favourable or less favourable. (ii) While N emissions to the environment need to be reduced for a number of reasons, controlling blooms can definitely be successful by reducing only P, provided concentrations of P can be brought down to levels sufficiently low to stringently limit biomass. Not the N:P ratio, but the absolute concentration of the limiting nutrient determines the maximum possible biomass of phytoplankton and thus of cyanobacteria. The absolute concentrations of N or P show which of the two nutrients is currently limiting biomass. N can be the nutrient of choice to reduce if achieving sufficiently low concentrations has chances of success. (iii) Where trophic and climate change cause longer, stronger and more frequent blooms, they increase risks of exposure, and health risks depend on the amount by which concentrations exceed those of current WHO cyanotoxin guideline values for the respective exposure situation. Where trophic change reduces phytoplankton biomass in the epilimnion, thus increasing transparency, cyanobacterial species composition may shift to those that reside on benthic surfaces or in the metalimnion, changing risks of exposure. We conclude that studying how environmental changes affect the genotype composition of cyanobacterial populations is a relatively new and exciting research field, holding promises for understanding the biological function of the wide range of metabolites found in cyanobacteria, of which only a small fraction is toxic to humans. Overall, management needs case-by-case assessments focusing on the impacts of environmental change on the respective waterbody, rather than generalisations.

**Keywords:** cyanobacteria; cyanobacterial toxins; climate change; eutrophication; health risk

## 1. Introduction

A large and increasing number of publications is voicing the concern that toxic cyanobacterial blooms will increase in consequence of climate change and eutrophication, and that this will substantially threaten human health [1–3]. In response to accelerated waterbody eutrophication in the second half of the 20th century, cyanobacterial blooms became increasingly common, and their detrimental effects on water quality moved into the focus of public and scientific attention. In parallel, research on phytoplankton moved from a descriptive botanical focus towards concepts for understanding drivers of the occurrence and dominance of species. For this, laboratory experiments on resource limitation differen-

tiated between uptake rates limited by concentrations of dissolved nutrients (i.e., Monod kinetics), rates of cell division limited by nutrient cell quotas (i.e., Droop kinetics) and total bioavailable nutrient fractions limiting the maximally attainable biomass in situ, leading to the concept of carrying capacity in its limnological interpretation [4]. This fed into the other important focus on phytoplankton populations in waterbodies: understanding the mechanisms of competition between species that determine occurrence and dominance.

The understanding of cyanobacterial toxins was pioneered by a small number of research groups in the 1960s and 1970s who elucidated the chemical structures of different toxin groups. By 1992, a variety of toxins was identified, in particular, hepatotoxic peptides (microcystins), neurotoxic alkaloids (anatoxins and saxitoxins), protein synthesis inhibiting alkaloids (cylindrospermopsins) and a neurotoxic organophosphate (anatoxin-a(S), syn. guanitoxin [5]). Cyanotoxin research gained momentum in the 1990s as the increasing availability of powerful methods of chemical analysis enabled wider surveys of cyanotoxin occurrence. This led to the recognition that the majority of blooms include globally occurring toxic species, and since the 1990s cyanobacterial blooms have been receiving massively increased research attention [6].

Cyanobacterial blooms continue to increase in response to eutrophication where agriculture is intensified or urban settlements expand without sufficient wastewater treatment. Elsewhere efforts to curb and reduce eutrophication began in the 1970s, focusing on phosphorus (P). Measures to reduce P emissions have increasingly been implemented, resulting in declining P concentrations in many waterbodies [7] and in some waterbodies eventually to less phytoplankton biomass and less cyanobacterial blooms [8]. Trophic change, i.e., rapid (anthropogenic) change in nutrient concentrations, is ongoing in both directions [9], eutrophication in many waterbodies and but also 'oligotrophication' in others.

Regarding cyanotoxins, we now have a general, albeit incomplete, picture of their occurrence in relation to cyanobacterial taxa, geography, and environmental conditions (reviewed in [10]). We also know that the toxicity of field populations primarily depends on their composition of toxic and non-toxic genotypes. Yet, our knowledge on conditions driving the occurrence of toxic genotypes is still sparse, and hence our understanding of the regulation of in situ toxin occurrence is quite limited.

Moreover, the currently known cyanotoxins are only a small fraction of (bio)chemically similar metabolites produced by cyanobacteria, and their function for the producing cells is yet unclear [11,12]. Toxicity to vertebrates is an evolutionary coincidence rather than an evolutionary adaptation. The implications of cyanotoxins for human health have been a key motivation for the study of cyanotoxins, and for the target of protecting public health this is undoubtedly justified. However, the focus on metabolites which happen to be toxic to humans but represent only a small fraction of the cyanobacterial metabolites hampers a better understanding of cyanobacterial biology and biochemistry.

More recently, a focus of research has been on understanding how trophic change and climate change act together to drive the occurrence of potentially toxic cyanobacteria, and the concern about health risks from their toxins has further promoted this line of research. As a result, some generalisations are increasingly being proposed, in particular, that climate change will exacerbate cyanobacterial blooms, that controlling blooms and toxin concentrations requires a balanced approach of reducing not only the concentrations of phosphorus (P) but also those of nitrogen (N), and also that the occurrence of cyanotoxins is a 'very serious' threat to human health.

In the following review, we address these generalisations, focusing on those concepts which are still subject to controversy and on newly arising concerns. We ask three questions and assess the respective evidence on the basis of both theoretical considerations and published information:

- In which types of waterbodies will climate change promote blooms (Section 2)?
- Can controlling blooms by focusing on the reduction of P loads be successful without also reducing N, and where eutrophication does decline, can this cause other risks by merely shifting cyanobacterial populations (Section 3)?

- How do trophic and climatic changes affect health risks from cyanobacteria (Section 4)?

For each of these points, we discuss how more recent views relate to earlier paradigms in limnology, proposing hypotheses for further discussion. Our aim is differentiation between well-substantiated generalisations and a need for situation-specific answers.

## 2. In Which Type of Waterbody Will Climate Change Promote Blooms?

Publications of the last two decades have increasingly addressed the potential for climate change, particularly warming, to influence cyanobacterial proliferation and blooms (reviewed, e.g., in [13,14]). While warming is frequently proposed to exacerbate cyanobacterial blooms, temperature itself is not a resource such as nutrients or light, and hence some important effects of changing temperatures on cyanobacterial blooms are indirect. The competitive advantage of cyanobacteria at high temperatures is not the only—and in many cases not the most relevant—response. Climate change impacts complex ecological systems in many ways, with some of the demonstrated and expected mechanisms causing conditions to be more favourable for cyanobacteria while others can have the opposite effect. In Section 2 we discuss both possibilities and attempt to narrow down the type of waterbody in which climate change is more likely to increase duration and/or intensity of cyanobacterial blooms.

### 2.1. Impact of Warming on Cyanobacteria and Microcystin Concentrations

Growth rates of cyanobacteria are generally lower than those of many eukaryotic phytoplankton species, and this is one of the mechanisms explaining why they often attain dominance only later in the growing season. The growth rates of some cyanobacterial taxa, particularly of *Microcystis*, show particularly steep curves of temperature-dependence: Coles and Jones [15] show this for three species of cyanobacteria relative to one diatom species. The relationships of growth rates to temperature presented by Reynolds [16] are still a popular reference for higher growth rates of cyanobacteria at higher temperature; however, they actually show a substantially steeper increase only for *Microcystis*, and at all temperatures the growth rates for cyanobacteria are lower than those for the six algal species (including three species of diatoms). The key message from this figure is that growth rates of these cyanobacteria still increase up to almost 30 °C (and for one strain even up to 40 °C) while for the eukaryotic algae they are generally higher, but increase only up to 20 °C. This is in line with the results of Butterwick et al. [17] who show that four out of six species of cyanobacteria still grow at 30 °C, while this was the case for only 6 out of 15 eukaryotic phytoplankton species. It is also in line with the results of Thomas and Litchman [18] who show optimum growth rates of 12 strains of *Microcystis*, *Raphidiopsis* (syn. *Cylindrospermopsis*) and *Dolichopsermum* (syn. *Anabaena*) in the range of 27–37 °C. Paerl and Otten [19] compile data from nine studies showing maximum growth rates of Cyanobacteria between 25 and 35 °C.

Regarding the more complex mechanisms determining species composition in the field, using microcosm experiments and modelling de Senerpont Domis et al. [20] show that although growth rates of cyanobacteria responded more strongly to warming than those of other species, this did not affect seasonal succession from diatoms over chlorophytes to cyanobacteria. Comparing growth rates of three strains of native cyanobacterial species to three strains of invasive species from warmer climates, Mehnert et al. [21] found that while the native ones can build populations earlier in the season, there is a potential for the invasive Nostocales, particularly *Chrysosporum* (syn. *Aphanizomenon*) *ovalisporum*, to outcompete them later in the season. Using the example of *Raphidiopsis raciborskii*, these authors add a further consideration: to unfold maximum growth rates at high temperatures, even shade tolerant species need high light intensity. In highly eutrophic and thus turbid waterbodies, this may counteract their competitive advantage given by high temperature.

A further direct impact of warming postulated by some authors is that higher temperatures favour microcystin-producing strains of *Microcystis* over non-producing ones. Lei et al. [22] discuss respective publications (see also Section 4.1), while for the results

of their own co-culture experiments they do not consider microcystin production the key mechanism leading to dominance. Ninio et al. [23] show that in laboratory experiments the *Microcystis* strain with a low MC content outcompetes the one with a high MC content particularly at high temperatures (20 and 25 °C), and this is in line with their field observation of the non-toxic strain increasingly dominating in Lake Kinneret in the wake of warming. These authors propose slight changes in temperature to be decisive for the outcome of competition between strains and suggest that there is no per se advantage of MC-producing over non-producing genotypes at elevated temperature; rather, this may vary between strains. Ninio et al. [23] conclude that "*this widely accepted model for the expansion and persistence of Microcystis blooms should be re-evaluated as higher temperatures may affect not just the intensity of the bloom, but also the composition of the Microcystis population, species, strains and genotypes*". As of now, a general claim of elevated temperatures favouring microcystin-producing cyanobacteria does not appear to be well evidenced by data.

### 2.2. Impact of Climate Change on Conditions in Waterbodies

Mechanisms through which overall climate change can shape the environmental conditions in waterbodies are manyfold and begin with the catchment. Heavy precipitation can cause nutrient input pulses from the catchment through soil erosion and run-off, but it can also dilute nutrient concentration in a waterbody. Conditions acting within a waterbody include impacts on duration and stability of thermal stratification, frequency of mixing events caused by storms, changes in external and internal nutrient loads and changes in photon flux density (via many mechanisms affecting mixing and turbidity). Cottingham et al. [24] emphasise the effect of changes in stratification patterns on the duration of cyanobacterial life stages in the pelagic versus the benthic zones and call for research to elucidate how these impact on population development. While elevated atmospheric $CO_2$ concentrations affect global temperatures, they can also influence the strain composition of cyanobacterial populations and thus toxin concentrations through increasing concentrations of dissolved $CO_2$ [25], particularly in eutrophic and hypertrophic waterbodies [26]. Shuvo et al. [27] comprehensively discuss the wide range of interacting variables that influence the extent of climate change effects. Besides temperature, this includes lake basin morphology and orientation, hydrology and trophic state but also solar irradiation and cloud cover.

For the direct impact of hydrophysical conditions, Reichwaldt and Ghadouani [28] give an overview of case reports in which rainfall events in lakes and reservoirs triggered an increase or decrease of cyanobacterial biomass. These authors quote increased turbidity and high turbulence as potential drivers of change—in some cases for a decrease and in others for an increase of blooms. They also emphasise the rather small number of studies published on the impact of rainfall patterns as well as the complexity of the mechanisms involved and list "*flushing, nutrient input, reduction of conductivity (especially after long dry periods), and mixing of the water column due to water inflow and associated strong winds*" as mechanisms by which rainfall events shape the environment to favour or disfavour cyanobacteria in phytoplankton communities. Huisman et al. [29] point to the relevance of time patterns of extreme events: where nutrient pulses from storms follow periods of drought, they are most likely to feed cyanobacterial proliferation if these are nutrient-limited. Yang et al. [30] show this with an 18-year time series of data from Lake Taihu, China, where high wind speeds and strong rainfall caused blooms to increase particularly since 2012, once nutrient concentrations had declined to levels potentially limiting biomass (i.e., TP < 80 µg/L, with TN still in the range of 1900–2400 µg/L). With a time series analysis from 2000 to 2014 for the large, deep pre-alpine Lago Maggiore, Morabito et al. [31] show that over longer time spans (months) pronounced rain events dilute rather than enrich nutrients; nonetheless within short time spans (5–7 days) cyanobacterial blooms showed some (albeit not significant) correlation to rainfall. Furthermore, soil characteristics in the catchment, such as iron content, have an impact on P loads following storm events, and

Tang et al. [32] conclude that an increase in extreme precipitation events may substantially mitigate eutrophication in some waterbodies while worsening the conditions in others.

Phytoplankton cells entrained in the turbulence of an epilimnion or shallow waterbody experience these hydrophysical changes in terms of exposure to light intensity and light–dark cycles and, where not only wind but also rainfall is involved, also in terms of nutrient pulses or dilution. The frequency at which waterbody conditions change can be an important consequence of climate change, as disturbances at intermediate time scales not only can disrupt cyanobacterial blooms, but generally increase phytoplankton diversity. Litchman et al. [33] analyse how the traits of different functional groups of phytoplankton determine their responses to these environmental changes and outline how the complex interplay of these mechanisms may be modelled. Anneville et al. [34] emphasise the challenge of studying the impact of extreme events in the field, as these vary considerably between years, possibly more so than the carrying capacity given by nutrient concentrations.

Publications on the impact of climate change on phytoplankton follow two different approaches: some evaluate data across a larger number of waterbodies for associations between blooms and elevated temperatures, applying multivariate statistics or neural networks [27,35,36]. Associations thus found can provide valuable indicative information particularly where different data sets and statistical approaches show similar results. However, in face of the different mechanisms acting in different types of waterbodies the results are sensitive to the choice of waterbodies and therefore may not well support generalisations. A smaller number of publications has studied time series of data from individual waterbodies to assess the impact of extreme events such as heat waves [30,34,37], extreme winters [38], rainfall patterns [28] or reduced mixing [39,40]. While many studies conclude that climate warming will exacerbate cyanobacterial blooms, some (discussed below) show the opposite. Key to understanding contradictory evidence is an understanding of the possible mechanisms and hierarchy of conditions determining the access of phytoplankton cells to nutrients and light. In the following, we propose that predictions about the impact of climate change are most reliable for nutrient-poor and for shallow waterbodies.

### 2.2.1. Climate Change Is Not Likely to Exacerbate Blooms in Oligotrophic and Oligo-Mesotrophic Waterbodies

Evaluating a large data set (2561 lakes across seven countries), Shuvo et al. [27] found TP to be the main single predictor of phytoplankton biomass (measured as Chl-*a* concentration), accounting for 42% of the variation, but spring and summer weather (temperature, precipitation and cloud cover) were almost equally relevant for the concentrations of Chl-*a*, while physical characteristics (residence time, mean depth, surface area and elevation) together accounted for 20% of the variation. However, this evaluation did not differentiate lakes by trophic status, and these percentages are specific to this data set. Where concentrations of the macronutrients N and P strongly limit total phytoplankton biomass, dominance of cyanobacteria is not likely and if it does occur, the low concentrations of N and P will not provide the capacity for the development of significant blooms (see Section 3.1.3). Rigosi et al. [35] analysed data of >1000 North American lakes classified by trophic state and show that in oligotrophic lakes cyanobacterial biomass related more strongly to nutrients, while in mesotrophic lakes temperature showed a stronger impact. In eutrophic and hyper-eutrophic lakes these authors observed a significant interaction between nutrients and temperature. Further, they found that whether nutrients or temperature—or both—promoted growth of cyanobacteria to be taxon-specific.

Elliott et al. [41] confirm this with a modelling study simulating a summer bloom of *Dolichospermum* which showed that while temperature promoted the growth of this taxon, the effect was much smaller than that of changes in the nutrient load. Using satellite data reflecting lake surface temperature and concentrations of Chl-*a* from 188 lakes, Kraemer et al. [42] found that in phytoplankton-rich lakes warmer conditions boosted Chl-*a* while this was not the case in phytoplankton-poor lakes. Gallina et al. [43] studied the impact of extreme weather events on 5 oligo- to moderately eutrophic, deep, stratified

peri-alpine lakes and found that warm summers did not lead to dominance of cyanobacteria: while their biomass increased during hot periods, their contribution to total phytoplankton increased only slightly, and the most striking response was a reduction in their species diversity. Anneville et al. [34] found that extremely warm summers and autumns did not lead to cyanobacterial blooms in oligotrophic alpine Lake Annecy whereas in the mesotrophic lakes Bourget and Geneva, they did increase, although not in summer but only in autumn. These authors also conclude that milder autumn temperatures benefit not only cyanobacteria, but other phytoplankton taxa as well, and this only if nutrient concentrations are sufficiently high. In shallow subtropical Peri Lake (southern Brazil) with rather low nutrient concentrations (5–20 µg/L TP), rainfall diluted TP rather than adding loading to the lake because import from land in this conserved watershed was low [44].

Climate change therefore is not likely to have much impact on bloom formation in oligotrophic waterbodies. Nonetheless, in very large oligotrophic waterbodies transient surface films can recruit themselves from the large water volume and, under certain conditions, accumulate to dense scums with potentially hazardous concentrations of cyanotoxins [45]. Further exceptions, discussed in Section 3.3, include cyanobacteria that grow on surfaces (including macrophytes), as well as metalimnetic layers in deep, stratified lakes, particularly *Planktotrhix rubescens*.

### 2.2.2. In Meso- and Eutrophic Waterbodies, Climate Change Can Act Both Ways: Exacerbating or Reducing Blooms

Where both P and N are ample or available beyond the amounts that phytoplankton can utilise to form more biomass, the impacts of climate change will act primarily through changing hydrophysical conditions. For example, in hypertrophic Nieuwe Meer with nutrient concentrations well above limiting phytoplankton biomass, the competitive advantage of *Microcystis* at high temperature and low mixing proved evident [37]. Mechanisms exacerbating blooms include the earlier onset of and prolongation of the growing season, giving cyanobacterial populations more time to outcompete other phytoplankton. Additionally, higher stability of stratification caused by earlier warming gives them a competitive advantage by allowing more effective vertical migration.

In contrast, however, pronounced stability of thermal stratification reduces upward diffusion of nutrients from the hypolimnion to the epilimnion and erosion of the metalimnion; it can trap nutrients in the hypolimnion as particles sink down from the epilimnion and are mineralised. Wentzky et al. [46] show this for Rappbode Reservoir, Germany: while the increase of the surface water temperature reflected that of air temperatures, the temperature in the hypolimnion showed a slight decrease, attributed to the significantly earlier onset and higher stability of thermal stratification. For Lake Garda, Salmaso et al. [40] show that warming induced incomplete mixing, thus causing "*climate warming-induced oligotrophication*" of the epilimnion, reducing the biomass of *Planktothrix*. Verburg et al. [47] describe a similar observation in tropical Lake Tanganyika, quoting reduced vertical mixing due to more pronounced water density gradients also for Lake Malawi.

Independently of trophic status, intermittent rainfall and storms, particularly if intensive, can disrupt blooms [28], re-setting seasonal species succession to an earlier stage. Using a 40-year data series, Posch et al. [39] show the pronounced impact of weather-induced variations in stratification on the biomass of *Planktothrix rubescens* in Lake Zürich.

### 2.2.3. Climate Change Is More Likely to Enhance Cyanobacterial Blooms in Shallow Than in Thermally Stratified Waterbodies

The impact of climate-driven hydrophysical conditions on the sediment–water interface tends to be more pronounced in shallow than in stably stratified waterbodies. One mechanism is phosphorus release from sediments: in shallow water warming reaches the sediment surface much more strongly, enhancing the mineralisation rate of organic matter and thus directly releasing P [48]. Hupfer et al. [49] show that more rapid mineralisation may, in turn, speed up the development of anoxic zones on the sediment surface, thus

promoting release of redox-sensitively adsorbed P (provided this is a significant fraction of P in the sediment).

A further mechanism is sediment resuspension by storms and heavy winds in shallow water, even more so if increased drought reduces their water level, and the mineralisation of detritus thus resuspended provides pulses of P which can feed blooms. Tammeorg et al. [50] demonstrate the relevance of stronger wind and wave action in Lake Peipsi, resuspending the seston that formed during the previous summer weeks which is readily biodegradable: it releases P that supplies late summer cyanobacterial population growth. In a modelling study for Lake Tegel, in which stratification is not very stable, Chorus and Schauser [51] show that storms occurring in late spring/early summer can prevent thermal stratification from developing at all. Thus, for the same lake, climate change can lead to more stable stratification in some years and destabilisation in others, depending on time patterns of warming and storms. Analysing data from 143 shallow lakes in Europe and South America, Kosten et al. [52] show that with increasing temperature the percentage that cyanobacteria contribute to total phytoplankton biomass increases steeply. These authors conclude that in a warmer climate controlling cyanobacterial dominance will require more pronounced nutrient reduction.

### 2.3. Conclusions for the Impact of Climate Change on Waterbody Conditions Favouring Blooms

A hierarchy of conditions determines the success of cyanobacteria: while nutrients "set the overall stage", weather events shape the more immediate reactions. They can enhance the frequency, duration and intensity of cyanobacterial blooms only within the frame given by the carrying capacity of resources for phytoplankton biomass, and in oligotrophic waterbodies this is low. This also applies to shallow waterbodies, provided they are oligotrophic, but due to the generally higher relevance of sediment–water contact, this is less likely to be the case. Thus, waterbody morphology can be seen as a secondary variable determining how climate change can affect the dominance and biomass of cyanobacteria.

Of course, there are exceptions to such a general pattern. Here we did not address the possible impacts of changes on food webs, in particular on grazing (discussed in Winder and Sommer [13] and Moustaka-Gouni and Sommer [53]) because of their relatively low relevance to cyanobacteria. However, Selmeczy et al. [54] describe the fascinating case of formerly oligotrophic, deep and stratified Lake Stechlin which is developing towards eutrophication without any recognisable external nutrient input, possibly because a change in hydrophysical conditions was induced by a climatic anomaly, leading to a chain reaction of shifts in the biota which in turn triggered release of P stored in the sediment.

The occasionally voiced perception of a "scientific consensus" that climate change will exacerbate cyanobacterial blooms is challenged by the diversity of waterbody conditions driving responses to change. A key conclusion that Richardson et al. [55] draw from their analysis of data from 494 lakes in central and northern Europe is the pronounced variability of cyanobacterial biomass relative to gradients in temperature, even though these authors differentiated their data by lake types (alkalinity, colour, mixing). Reichwaldt and Ghadouani [28] expect a strong impact of climate change but emphasise that it will be "*very complex and will strongly depend on site-specific dynamics, cyanobacterial species composition, and cyanobacterial strain succession*". Furthermore, Mooij et al. [56] warn that the "*reuse of eutrophication models for studying climate change is a logical step but should be done with great care, because the validity of the outcomes has generally not yet been properly tested against empirical data, and field studies show clear synergistic effects that are not well covered by existing models.*" From the research reviewed above we share the view that while the evidence base for generalisations on the impact of climate change is still limited, for the target of avoiding cyanobacterial blooms it is clearly prudent to strengthen the efforts to mitigate eutrophication.

### 3. Concepts for Mitigating Eutrophication—Do They Need to Include N, and Can Reducing Trophic State Risk Shifting Cyanobacteria to Metalimnetic or Benthic Habitats?

Based on Liebig's law of the limiting nutrient and the recognition of the relevance of P in this role by Vollenweider [57] and others, many of the efforts to mitigate eutrophication of freshwaters have strongly focused on reducing P loads. This is also due to P being easier to control: banning it from detergents almost halved loads from some domestic wastewaters within a few months [58,59]. However, in most situations this alone is insufficient for controlling eutrophication, and implementing P removal in sewage treatment, at minimum through simultaneous precipitation, is also necessary [60]. While efficient P removal is widely implemented [61], cost-effective and efficient procedures for N removal are not yet equally established, largely for technical and economic reasons [62,63]. For coastal waters the prevalence of N-limitation was recognised, and some regulations also address N loads (e.g., those of the European Union and some intergovernmental river-basin management treaties); however, generally, these efforts have been less successful: where load reduction has occurred, it is substantially more pronounced for P than for N, particularly for loads from agriculture (reviewed by Glibert et al. [64]). These authors also show that in regions with rapidly developing and growing economies, N loads are increasing dramatically, and that urea is increasingly replacing the ammonium-nitrate fertilisers which are more challenging to handle because of being explosive. As cyanobacteria more readily use reduced forms of N, this can promote their proliferation [65].

The more recent awareness of the relevance of forms of nitrogen for promoting cyanobacterial blooms [66] has led to a strong focus on the ratio of N:P in numerous publications, with high N:P-ratios being interpreted as indicating a need to also reduce the concentrations of N in the respective waterbody and often without explicit attention to the absolute concentrations of either nutrient. This also includes experiments adding nutrients to field samples, from which the same conclusion is frequently drawn if adding not only P but also N enhances growth more than adding only N or only P. These approaches appear to be in conflict with the state of knowledge on the mechanisms of nutrient limitation established in the last decades as well as with the increasing amount of data showing substantial decline of phytoplankton biomass, particularly that of cyanobacteria, once P concentrations undercut threshold concentrations. Clarity on the most effective approaches to bloom control is, however, important for implementing measures to curb the biomass they can attain in the respective waterbody. This includes the need to explicitly differentiate between the limitation of nutrient uptake rates (which can limit growth rates) by dissolved N and P, and the limitation of biomass (sometimes referred to as 'yield' or 'standing stock') by total N and P. The widely used term 'growth limitation' is unclear and can imply either or both. It is also important to understand whether or not measures in an upstream waterbody are likely to '*displace eutrophication downstream*' [67] and where a 'dual strategy', addressing both nutrients [14,19], is the more effective way forward.

In this section we discuss the conclusions drawn from N:P ratios and nutrient addition experiments relative to the state of knowledge established in the last century. From this we draw conclusions for aspects that a 'dual strategy' needs to consider in order to develop a differentiated approach, based on case-by-case assessments.

#### 3.1. N:P Ratios and Growth Responses to Adding Both Nutrients Relative to the Absolute Concentrations of N and P

Particularly during the 1980s and 1990s, research in phytoplankton ecology focused on elucidating the mechanisms of nutrient limitation. A key outcome was the paradigm that not the N:P ratio, but the absolute concentration of the limiting nutrient determines the maximum possible biomass of phytoplankton and thus of cyanobacteria. It follows that whether N or P are currently limiting biomass can be inferred from their absolute concentrations. We postulate that this paradigm still holds true, for the reasons discussed in Sections 3.1.1–3.1.6. While shifting N:P ratios certainly can have downstream effects,

we also postulate that these require case-by-case assessments rather than a generic 'dual strategy' (Section 3.1.6).

### 3.1.1. Evidence Regarding Dissolved Nutrient Fractions

Phytoplankton cells deplete of P, N or both will quickly incorporate dissolved nutrients whenever they become available and use them for building new biomass (reviewed in Salmaso and Tolotti [68]). If dissolved P is available in excess of the current need for cell division, it can be stored as polyphosphate for up to four further divisions [69]. Cyanobacteria are also capable of storing excess N in phycobiliproteins and, in particular, in cyanophycin granules [70]. The latter can constitute up to 18% of cellular dry weight [71]. Their formation is triggered by depletion of other resources (P, light or sulfate), and they are catabolised to enter the nitrogen metabolism when other resources no longer limit growth [72]. Further, a number of bloom-forming cyanobacterial taxa can access the atmospheric N-pool through diazotrophy [73]. However, in turbid, eutrophic waterbodies N-fixation rarely contributes significantly to compensating N-limitation because this process is energy-intensive [74,75].

To some extent cyanobacterial cells can up- or downregulate their uptake rate of dissolved P and do this in particular in response to pulses of dissolved nutrients (see Cáceres et al. [76], on P transporters). Nonetheless, uptake rates are limited by the cells' nutrient uptake mechanisms, and above a saturation concentration they cannot further increase uptake rates. However, species differ in their nutrient uptake kinetics, generally characterised by maximum uptake rates and half-saturation constants, and these differences are important for the outcome of competition between species under nutrient limiting conditions or fluctuating nutrient concentrations [77]. Concentrations above which uptake rates reach saturation can be assumed for all of the phytoplankton species and are well established for a number of taxa. Generally, there appear to be no species for which uptake rates are not saturated if DIN is >100–130 µg/L and DIP is >3–10 µg/L [78–81]. In consequence, nutrient limitation is given only in situations in which the concentrations of dissolved nutrients are below these saturating levels.

### 3.1.2. Evidence Regarding Total Nutrient Fractions

There is an upper limit to the maximally attainable biomass that can be sustained by the ambient resource at a given point in time, established in phytoplankton ecology since the 1990s as "carrying capacity". Using the example of Rostherne Mere, Reynolds and Bellinger [82] explains how resources alternate over the course of a year to set this upper limit: in temperate climates this is typically light during winter, and in oligotrophic to slightly eutrophic waterbodies limitation may shift to total P and/or total N during summer, once all dissolved nutrient is incorporated into biomass and detritus. At high nutrient concentrations cell density may reach a point at which it causes such pronounced turbidity that light availability limits any further increase of phytoplankton biomass [83]. Accordingly, a range of studies has shown threshold concentrations of TP above which higher concentrations do not lead to more biomass. Some of these studies compare annual or seasonal means between many lakes [68,78,84–86]. Others follow the trophic development of individual lakes (see case studies in Fastner et al. [87] and Salonen et al. [88]). Taken together, these data show that in the plots of biomass versus TP concentrations, biomass typically levels off at 50–100 µg/L TP (see also Quinlan et al. [89]). Of course, in multi-lake studies the scatter is considerable, due to further specific conditions constraining growth such as background turbidity or losses through grazing and sedimentation. In particular, the stability and depth of thermal stratification influences light limitation: the TP-threshold at which biomass levels off can be higher in very shallow waterbodies and lower in those with a deeply mixed epilimnion.

A TN-threshold concentration below which N is likely to limit further biomass increase can be tentatively inferred from the threshold concentration for TP by applying the Redfield Ratio of 16:1 as atomic ratio of N:P or as mass ratio of 7:1, as first introduced by Redfield [90,91] from observations on marine phytoplankton. This ratio is still widely used,

although it is only an average, and more recent work, including on freshwater phytoplankton, has shown that atomic ratios of N:P range up to 45:1 (mass ratio 20:1), in extreme cases even up to 133:1 (mass ratio 60:1) for the biomass of some (marine) species [92]. Thus, based on a TP-threshold of 50 μg/L, a TN threshold would amount to 350 μg/L when using the Redfield Ratio and to 2000 μg/L using a TP-threshold of 100 μg/L and a mass ratio of N:P = 20:1. This leads to TN concentrations above which N-limitation of phytoplankton biomass becomes unlikely in the range of 350–2000 μg/L. In their evaluation of 102 lakes in northern Germany, Dolman et al. [78] found biovolumes plotted against TN did not level off up to a TN concentration of 2000 μg/L, while for more than 800 Danish lakes Søndergaard et al. [93] found that above 500 μg/L TN, biomass was strongly dependent on TP and scarcely influenced by TN.

### 3.1.3. Evidence Regarding Cyanobacterial Blooms

TP-concentrations indirectly influence species composition because they limit the maximum phytoplankton biomass and thus turbidity: where water stays clear because low TP-concentrations cannot support a high amount of biomass, species that achieve a high growth rate at high light intensity tend to outcompete cyanobacteria. This was first demonstrated experimentally by Mur et al. [94] with the example of *Scenedesmus quadricauda* and *Planktothrix agardhii* and has been widely confirmed with field data showing that cyanobacteria tend to reach high levels of biomass and to dominate the phytoplankton where nutrient concentrations and thus biomass are high. Relative to nutrient concentrations, field data also show similar patterns for cyanobacterial biomass or dominance: Downing et al. [95] conducted a meta-analysis covering 99 temperate lakes worldwide and show that the summer mean percentage contribution of cyanobacteria to total phytoplankton biomass increases with the concentration of TP "*from a minimal fraction in nutrient-poor oligotrophic lakes to an asymptotic average of 60% above total P of 80–90 μg/L*". Other field studies on large data sets followed, albeit chiefly on waterbodies in temperate climates: Jeppesen et al. [96] evaluated 35 studies of trophic recovery encompassing a range of lake types, differentiating between shallow and stratified lakes; these authors found a substantial contribution of cyanobacteria to total phytoplankton biovolume above a threshold of 50 μg/L TP for shallow lakes and occasionally as low as 10–15 μg/L in stratified lakes. Carvalho et al. [86] analysed summer mean phytoplankton data from >1500 lakes in 16 countries reported under the EU Water Framework Directive and found that in the TP concentration range of 10–100 μg/L cyanobacterial biomass increased, but the relationship flattened at higher TP concentrations. Vuorio et al. [97] analysed data from 5678 phytoplankton samples from 2029 boreal lakes, differentiating between genera and water colour to determine TP thresholds at which regressions of biomass versus TP concentrations level off; these ranged from 10 μg/L for *Planktothrix* in oligohumic lakes to 50 μg/L for *Microcystis* in polyhumic lakes. Salmaso and Tolotti [68] review a number of studies, largely from temperate climates, and conclude an "*increasing dominance of cyanobacteria with TP increase*", with cyanobacteria typically dominating at higher total biovolume levels: they contributed >50% if total biovolume was >3 mm³/L and >75% if total biovolume was >40 mm³/L.

Field data on cyanobacterial dominance for other climates are scarce. For 14 tropical reservoirs in Venezuela, González and Quirós [98] confirm the pattern observed in temperate climates: their data show dominance of cyanobacteria in 8 of the 10 thermally stratified reservoirs in which TP concentrations exceeded 18–20 μg/L, and the only two exceptions were lakes with residence times of only 12 days. Such hydraulic conditions that keep the system in a constant early stage of plankton succession where populations of cyanobacteria simply do not have the time to develop [99]. Further, although the TN concentrations in this study correlated closely with those of TP, the atomic N:P ratios above 12 suggests that phytoplankton biomass in all of these lakes was not limited by nitrogen.

### 3.1.4. Misinterpretation of the Role of N and of N:P Ratios

Smith [100] published the first evaluation of cyanobacterial dominance in relation to N:P ratios, and this is often quoted to argue a need to assess limitation using N:P ratios. Downing et al. [95] discussed the contemporary controversy about the relevance of N:P ratios for cyanobacterial dominance, quoting Trimbee and Prepas [101] who added further data to Smith's assessment and concluded that the absolute concentrations of TP and TN were better predictors of cyanobacterial dominance than the ratio of TN:TP. Indeed, plots of biomass data (for total phytoplankton or for cyanobacteria) against TN or the ratio of TN:TP often show patterns similar to those of plots of biomass against TP, with similar or even higher correlation coefficients [95,98] or close associations in multivariate statistical analyses such as PCA. Such statistical results are sometimes interpreted as indicating co-limitation. However, a key mechanism causing such statistical outcomes is that the loads of both nutrients typically increase in parallel [95,102], reflecting their shared origin from sewage, mineral fertilisers or manure. In consequence, biomass can correlate with N or N:P even if the levels of N are far too high to be limiting, and actually P is the limiting nutrient. Hence, as basic rule, in nutrient-saturated situations N:P ratios are meaningless. If N:P ratios are applied to assess which of the two nutrients may currently be limiting it is therefore important to report whether data above a threshold concentration were excluded.

### 3.1.5. Co-Limitation and the Misinterpretation of Enhanced Growth in Experiments Adding Both N and P

Results of experiments at different scales (from micro- over mesocosms to whole lake) adding N and P separately as well as in combination have been widely published, and often they show the highest biomass increase when adding both N and P as compared to that when adding only one nutrient [75]. For two reasons such a growth response does not, however, imply that both nutrients are limiting phytoplankton biomass in the field. A straightforward reason is that light as further growth-limiting resource has rarely been adequately controlled in such experiments while in highly eutrophic waterbodies it is often the limiting resource. The other reason is that even if adding both nutrients leads to more biomass, the inverse does not apply: if one of the two nutrients is minimised, the amount of biomass that can form cannot exceed the amount supported by that nutrient (Figure 1). For field situations, this not only requires the load to be reduced, but that water exchange also leads to an export of the respective nutrient (including of the plankton and detritus containing it) from the waterbody.

Where the concentrations of both dissolved, bioavailable N and P are low enough to limit uptake rates (as discussed above: <3–10 μg/L for DIP and <100–130 μg/L for DIN), limitation may oscillate between N and P as pulses of nutrients reach the waterbody externally or from recycling processes such as biomass degradation and as phytoplankton uptake draws down their concentration. This is, however, not a co-limitation by N and P in a strict sense but rather a sequential shift in limiting factors. This mechanism has been termed sequential co-limitation by Saito et al. [103] or serial co-limitation by Harpole et al. [104] and is in agreement with Liebig's Law of one nutrient limiting at a time.

Liebig's law, developed for fertilising agriculture to increase the yield of crops in the mid 19th century, has a long history of debate on its applicability in aquatic ecology. De Baar [105] traced this with a pronounced emphasis on the limitation of growth processes. Dolman and Wiedner [106] show that when applying Liebig's law not to processes but to biomass—or to use agricultural terminology, to yield or standing stock—it holds: from analysing three large data sets from Germany and the USA, these authors "*find that indeed phytoplankton biomass, measured across a set of lakes, is better predicted separately from either TN or TP according to whichever is relatively least available . . . than from either TP alone . . . or jointly using both TP and TN for each lake*". They emphasise the decisive role of cell-internal nutrient concentrations (cell quota) for growth and propose a role of nutrient ratios, averaged over time, to indicate "*which nutrient is most likely to have the lower cell quota*".

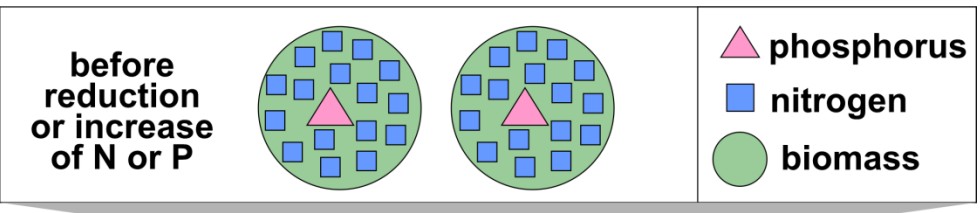

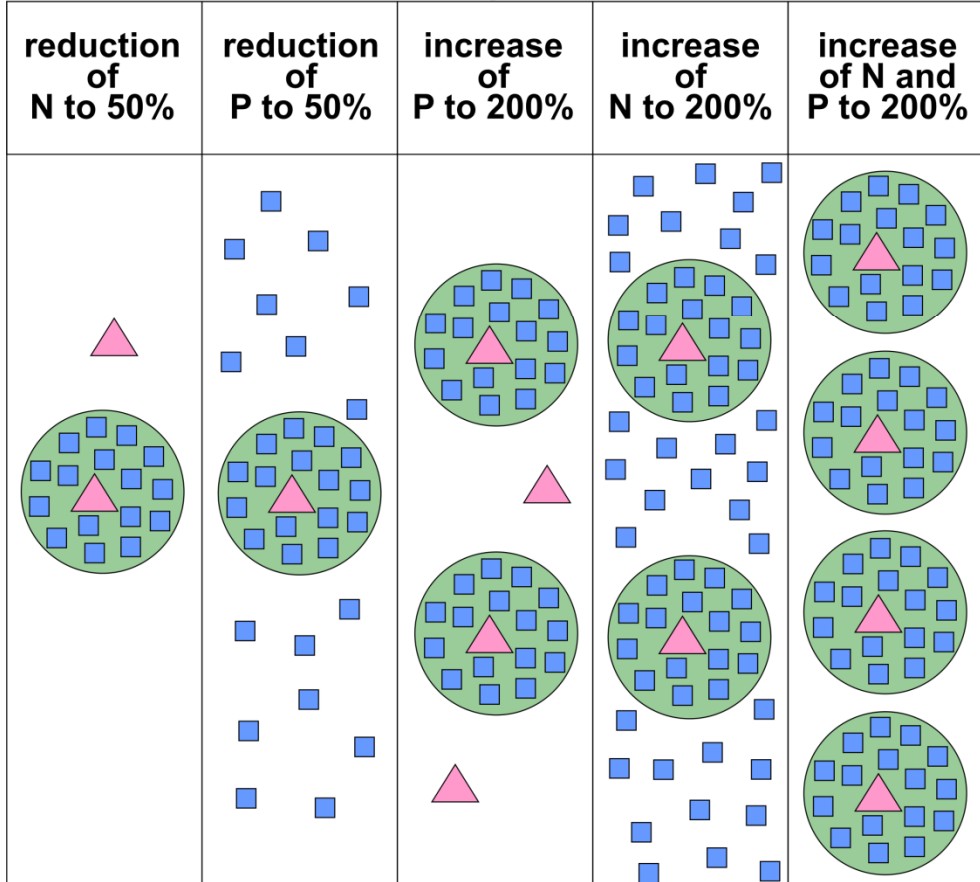

**Figure 1.** Schematic illustration of the impact of increasing or reducing nitrogen (N, blue squares) and phosphorus (P, pink triangles) concentrations, either individually or both, in a moderately eutrophic situation in which cellular uptake has depleted dissolved N and P and both nutrients are incorporated in cells (green circles). Nutrients that are not incorporated into biomass remain in the dissolved fraction outside of cells.

Why then conduct nutrient addition experiments? For management decisions, identifying sequential or serial co-limitation is not relevant because the timescale of fluctuating limitation is much smaller (days to weeks) than the timescale for planning and implementing policies and technical measures (years to decades). Restoration measures are most likely to be effective if they address the nutrient that has the highest probability of effectively reducing total phytoplankton biomass and thus cyanobacterial blooms, regardless of which nutrient is currently limiting [107].

Investigating nutrient limitation in field populations is relevant for understanding species dominance, as differences between phytoplankton taxa in nutrient uptake rates are decisive for the outcome of competition between them. A worthwhile scientific target for experiments with adding nutrients to field populations in micro- or mesocosm therefore is to elucidate which taxa are the superior competitors for the limiting resource(s) and end up dominating, both under stringent nutrient limitation and in response to additions

of N, P or both, or to investigate how pulsed nutrient addition can promote species' diversity (pioneered, e.g., by Tilman et al. [108] and Sommer [109]). Such experiments are particularly valuable if their design includes assessing light limitation, as done, e.g., by Maberly et al. [110] and Kolzau et al. [79]. This target requires differentiating the growth rates observed in the experiments by species or genera, but most published nutrient addition experiments are limited to a bulk biomass parameter, usually chlorophyll-a, and differentiation by taxa, as done by Müller and Mitrovic [111] is rare.

In consequence, nutrient addition experiments can provide insights on the drivers of species composition—including cyanobacterial dominance or even the outcome of competition between cyanotoxin-producing genotypes and non-producers (see below). For management decisions on the nutrient to reduce for the target of controlling cyanobacterial blooms, neither N:P ratios nor nutrient enrichment experiments add insights. To control trophic state and curb blooms it is scarcely relevant to determine which nutrient is currently limiting phytoplankton biomass: sufficiently reducing one nutrient will render that one limiting, regardless of which one was limiting before load reduction [107]. The key criterion is that sufficient reduction is feasible.

### 3.1.6. Criteria for the Choice of N or P to Control Blooms

There is no doubt that for overall environmental protection it is desirable to reduce the loads of both excess N and P to the environment (see below). Nevertheless, for the target of mitigating cyanobacterial blooms of health-relevant dimensions in a specific waterbody, setting a priority on N or P may be necessary. A key criterion for the specific waterbody is a practical one, i.e., the feasibility of achieving a sufficiently strong reduction to limit biomass. For P this has been achieved in an increasing number of cases, particularly through controlling point sources from wastewater (see above). While adding the necessary steps to sewage treatment has become feasible for controlling relevant point sources of P, reducing N loads from sewage may be more costly [112], although techniques are increasingly being developed and established [113].

Controlling nutrient loads from non-point sources, in particular from agriculture, is generally more challenging [14], as this requires collaboration with and compliance of a larger number of stakeholders, often with opposing interests, in the catchment.

### 3.2. Criteria for Addressing P or N and for a 'Dual Strategy' Addressing Both

One argument for the need of a dual strategy of reducing P and N is failure to sufficiently reduce external P loads. However, where restoration by P load reduction has not been successful, the key question (discussed in Section 3.2.1) is whether there are chances for a sufficiently pronounced reduction of the N load. A further argument is that internal P loads prevent TP from reaching target concentrations that effectively curb biomass, i.e., P cycled internally between water and anaerobic sediments. The question is, however, whether this internal load is indeed chiefly from redox-sensitive release of legacy P, or from mineralisation of freshly sedimented organic matter. Hupfer and Lewandowski [49] pick up on the paradigm that the former precludes restoration success: "*The apparent correlation between hypolimnetic oxygen depletion and hypolimnetic P accumulation as two typical symptoms of increased eutrophication in thermally stratified lakes was misinterpreted by reducing the complex processes at the SWI* (sediment-water interface) *to the statement 'oxygen depletion leads to release of phosphorus', which subsequently has become the accepted paradigm of the day*". These authors show that in many cases mineralisation is actually the major source of internal loading. This has important cause–effect implications because once external loading decreases to levels effectively reducing primary production, there is less organic matter to sediment and decompose, releasing P. This is why internal measures to intercept P release, such as sediment capping or treatment, need to be periodically repeated if the external load is not sufficiently reduced. Before concluding that P cannot be sufficiently reduced because of high internal loading it is therefore important to understand

the mechanisms causing P release and the relevance of redox conditions for this process (Section 3.2.1).

A further aspect discussed by Conley et al. [67] is that where P control is successful and biomass substantially declines, this can lead to more N being 'left over' and discharged from the waterbody. In consequence, reducing only P may result in more N reaching downstream aquatic ecosystems. However, whether or not this is the case and if so, whether or not it triggers adverse effects critically depends on conditions in the respective downstream waterbodies (Section 3.2.2). Both aspects require careful case-by-case assessments (Section 3.2.3).

### 3.2.1. Impacts of N within the Waterbody

In contrast to P load reduction, success stories are scarcely known from load reduction measures focusing on N [114]. One is the case of Müggelsee in Berlin, Germany, which is shallow and subject to substantial internal P loads from the sediment during summer as a legacy of decades of intensive agriculture upstream. As a result of restructured agriculture following the German reunification, fertiliser input declined, leading to substantial N-load reduction and, through in-lake denitrification, to phases of about 100 days of N-limitation during late summer. In consequence, cyanobacterial biomass declined by 89% compared to the average during the 1980s, interestingly without a shift to diazotrophic taxa such as *Aphanizomenon* sp. [114]. This demonstrates that substantial N load reduction can lead to N concentrations in the lake which are sufficiently low to effectively limit cyanobacterial blooms. In contrast, the simultaneous reduction of the P load to this lake was compensated by high internal loads which are likely to remain high for some years to come, i.e., until water exchange has exported sufficient amounts of phosphorus out of the system.

Candidate waterbodies for a focus on N include those in which N is already limiting or concentrations are not far above a potentially limiting level, thus rendering further reduction is a realistic option, as is typical for many shallow waterbodies during late summer. As one possible option of a fairly low-cost measure with the potential for a strong impact in such situations, Kolzau et al. [79] propose agreements with farmers in the catchment to time their application of fertilisers or manure so that run-off and leaching from agricultural soils is minimised during the cyanobacterial growth season. Chorus and Spijkerman [75] discuss the perspectives of controlling eutrophication by reducing N loads in some detail.

In some situations, in which internal P loading due to release of redox-sensitively bound P from sediments is a relevant source of P, maintaining some nitrate in the water above the sediment may even be beneficial. Steinman and Spears [115] emphasise the coupling of internal processes of N and P cycling through the role of nitrate for redox-sensitive P release mentioned above, and hypolimnetic concentrations of nitrate N > 0.5 mg/L are well known to effectively suppress the release of P from sediments by maintaining a sufficiently high redox potential at the sediment surface. Andersen [116] showed this with data from nine Danish lakes in which, even in anoxic hypolimnia, no P release was observed if nitrate N concentrations were in the range of 1 mg/L. For lakes without thermal stratification he reports a nitrate-N threshold of 0.5 mg/L below which P release may occur even if the water was '*well oxygenated*', possibly as nitrate transports oxygen into the interstitial water very effectively [117]. Salonen et al. [88] show pronounced hyperbolic curves for the concentrations of both TP and TN in the sediment-near water of Lake Vesijärvi relative to the concentration of nitrate-N: the concentrations of both TP and TN increased massively once nitrate was depleted. Schauser et al. [118] also discuss this for Lake Tegel. The case studies for both these lakes, however, confirm the above-mentioned assessment by Hupfer and Lewandowski [49] that redox-sensitive release may be a minor source of P relative to mineralisation of P bound in the large amounts of biomass formed in the lake on the basis of external P sources.

The overall ecological assessment of purposely adding nitrate products to suppress internal P loading [119] may be questionable (perhaps unless its consumption through

denitrification can be demonstrated). The available experience does, however, show that making use of nitrate for sediment oxidation can be an option, provided there is a good understanding of the respective sediment–water system and its nutrient budgets. In particular, before counting on nitrate to suppress internal P loading it is important to assess the relevance of redox-sensitive P release relative to that through mineralisation.

### 3.2.2. Downstream Impacts of N

In waterbodies previously N-limited during summer, reducing biomass by reducing TP concentrations can lead to dissolved N then being 'left over' by the phytoplankton; also, less biomass production leads to less organic substance available to fuel denitrification. In consequence, more N is exported to downstream ecosystems. This concern is addressed by Paerl et al. [120] with the example of the Neuse River (and further studies quoted therein) where upstream P reduction led the increase of downstream phytoplankton biomass in previously N-limited estuarine and coastal waters. An important criterion for a 'dual strategy' therefore is possible downstream impacts of an imbalance in the reduction of N and P.

Ample evidence shows that with some exceptions most coastal and estuarine waters are currently N-limited (see reviews by Howarth and Marino [121] or Malone and Newton [122]). However, as discussed above for limnetic systems, reducing P loads to sufficiently low levels can switch limitation to P also for coastal waters, particularly where water exchange rates are high and thus dilute or export P, including that adsorbed to sediments which is resuspended during strong wind events and storms. In some coastal settings the success chances for management measures to achieve stringent nutrient limitation may be higher for P than for N merely because of the dramatic increase of N loads relative to those of P. Reviewing N loads to coastal areas, Malone and Newton [122] quote projections for 2050 that propose a doubling of global N input to the oceans as compared to 1990 due to increased synthetic fertiliser use (particularly in Asia). Statham [123] shows atmospheric deposition to be an important source, accounting for 15–50% of the N loads to the estuaries studied. The regulations and management measures established in many countries during the past decades that were effective mostly for reducing P loads are showing success also for some coastal waters, as discussed by Howarth and Marino [121] for the North Sea estuaries and Laholm Bay in Sweden.

A strong argument for eutrophication control through limiting N loads is that denitrification can permanently remove N from aquatic systems to the atmosphere as $N_2$. This applies to freshwater as well as to marine systems: the share of N that is transformed to $N_2$ is estimated to be 40–50% for the northern Gulf of Mexico, 25% for the Great Barrier Reef, 42–96% for the Baltic Sea, 40–42% for the Northern Adriatic Sea and 40% for the Chesapeake Bay [122]. Phosphorus can also be effectively lost through burial in the sediment: in low salinity sediments a relevant fraction of P may be permanently bound in insoluble vivianite [124]. In shallow waters subject to sediment being resuspended, P adsorbed to sediment particles can be remobilised through wind-driven turbulence. Nonetheless, in their review of nutrient retention in the coastal areas of the Baltic Sea Carstensen et al. [124] show that in some estuarine regions removal of P by burial in sediments is more pronounced than losses of N, both through burial of organic matter and through denitrification. Losses of both N and P from the euphotic coastal zones also strongly depend on water retention times, this is, transport to the open ocean.

Thus, increased N concentrations in effluents of a waterbody undergoing oligotrophication through P-load reduction may or may not lead to increased downstream eutrophication, strongly depending on conditions in the downstream waterbodies: if P is already limiting in these, some additional N should not have an effect. If N concentrations are far above any chance of limiting biomass, slightly augmenting them through upstream oligotrophication should also have no effect. Furthermore, it should have no effect if the effluent of the upstream waterbody flows through wetlands with massive denitrification. It is therefore important that site-specific assessments are conducted at a larger scale, taking

into account the downstream impact of measures intended to control blooms in a specific upstream waterbody.

### 3.2.3. Case-by-Case Assessments

While this may take some years, the increasing experience with oligotrophication shows that both for lakes and coastal areas, later on in the recovery process sediments can again become a sink rather than a source for P [8,125], provided that the external load is sufficiently reduced and that water residence times are not too high. In conclusion of compiling data from a large number of lakes, Steinman and Spears [115] show that sediment TP concentrations of many lakes are quite similar across a wide trophic spectrum: they are elevated only in some eutrophic lakes. These authors strongly emphasise the need for "*lake-specific analyses to determine the best management strategy; a one-size-fits-all approach is doomed to failure*". Lewis and Wurtsbaugh [107] comprehensively discuss situations in which N-limitation of total phytoplankton biomass is likely, they emphasise that in most cases restriction of P loads is the "*most promising management tool*" while also acknowledging the role of N-limitation in phosphorus-saturated waterbodies. Already more than 10 years ago, these authors pointed out the need to reduce atmospheric N deposition to prevent currently N-limited lakes from becoming more eutrophic.

Taken together, the current state of knowledge shows that limiting only one nutrient is highly effective if it is brought down to sufficiently low concentrations, both for freshwater and marine systems. However, whether or not such target concentrations can be achieved within the desired timeframe and whether or not negative downstream impacts are likely depends on the specific conditions. Generally requiring the reduction of the loads of both nutrients can dilute the available funding and by this increase the risk that neither measure for N nor P reduction is sufficiently efficient to bring concentrations down low enough to sustainably limit phytoplankton biomass.

We therefore propose to supplement the call for a 'dual strategy' with a call for basing this on case-by-case assessments that include not only an assessment of the technical, political and societal feasibility of attaining sufficiently low target concentrations, but also of impacts on downstream ecosystems.

### 3.3. Shifts in Cyanobacterial Populations in Consequence of Oligotrophication

A specific aspect of trophic change is the reduction of trophic state leading to more transparent water. This (re-)opens habitats for metalimnetic, tychoplanktonic and benthic cyanobacteria. In Section 4.3 we discuss how the challenges these present for human health risk assessment differ from those of epilimnetic cyanobacteria.

### 3.3.1. Disappearance and Re-Appearance of *Planktothrix rubescens* in the Metalimnion

*Planktotrhix rubescens* is the most prominent and best studied example. Most strains of this species produce microcystins [126,127] and dense populations typically occur in meso-eutrophic deep, stratified lakes primarily in the metalimnion (mostly between 10 and 20 m) [128]. This low-light adapted species can effectively use the very low intensity of green light in these depths as well as adapt to variable light conditions by means of buoyancy regulation [128–130].

The occurrence of *P. rubescens* is closely linked to trophic state. It appeared in many formerly oligotrophic deep lakes such as Lake Mondsee, Lake Zürich and other peri- and subalpine lakes as they shifted from oligotrophic to mesotrophic [131–133]. With further eutrophication, however, *P. rubescens* disappeared due lack of light in the metalimnion caused by high epilimnetic biomass [134]. In times of highest trophic state of these lakes, other cyanobacteria such as *Microcystis*, *Aphanizomenon* and *Dolichospermum*, but also green algae or diatoms were reported as dominant taxa in the epilimnion (e.g., Chiemsee [135], Lake Zürich [131], Lake Baldegg [133], Lac du Bourget [136]). As successful management measures to reduce trophic state led to a return of higher transparency, epilimnetic

cyanobacteria disappeared and *P. rubescens* (re-)appeared in many of these lakes, a process termed "paradoxical outcome" of lake restoration by Jacquet et al. [129].

*P. rubescens* can thrive at temporarily lower P-concentrations because, like other cyanobacteria, it can effectively compensate the lower phosphorus concentrations in the epilimnion by storing P well in excess of the current need and/or by excreting alkaline phosphatase [137]. However, primarily it benefits from the usually higher nutrient concentrations in the metalimnion compared to the epilimnion. In the phase of the decline of TP concentrations *P. rubescens* attained biovolumes up to ~10 mm$^3$/L in the metalimnion and dominated the total phytoplankton biomass. This changed once total phosphorus (TP) declined below 10 μg/L and in consequence *P. rubescens* declined to biovolumes <1 mm$^3$/L or even was absent [128,132,136,138].

During winter mixing of waterbodies, *P. rubescens* can occur dispersed over the entire water column or, once mixing subsides, it can even form dense surface blooms rendering the water deeply reddish-brown [139] and in which microcystin concentrations can range up to >30 mg/L [140,141]. In the metalimnion microcystin concentrations are generally much lower with reported values rarely exceeding 10 μg/L [142].

Where TP concentrations are around or slightly above 10 μg/L the abundance of *P. rubescens* can vary substantially between years, including periodic absence as, e.g., in Lac du Bourget and Ammersee [136,143]. Such annual variability has been explained by fluctuations of water transparency and thus of ratios of euphotic depths to mixing depths $Z_{eu}:Z_{mix}$. In contrast, constantly elevated biomass of *P. rubescens* is attributed to changes in the patterns of stratification and mixing due to climate change. During the last decades, years with incomplete mixing have become more frequent, for example, in Lake Zürich, allowing *P. rubescens* to increase its population as more cells survive as inoculum for the next season's population while holomixis would diminish cell density more strongly due to higher hydrostatic pressure in deep water layers [39,144]. On the other hand, reduced mixing can also result in nutrient depletion of the epilimnion through sedimentation and thus lead to lower abundance of *P. rubescens*, as observed in Lake Garda [40]. Higher temperatures early in the year and thus an earlier onset of stratification have been found to favour *P. rubescens* in Mondsee [128] and Lake Geneva [145].

The current state of knowledge of conditions conducive to *P. rubescens* shows that in many deep, thermally stratified lakes and reservoirs the combined effects of changing nutrient concentrations and climate-driven changes in stratification patterns can work both ways, this is, either favour or disfavour the proliferation of this species. For water body management and use—especially for the abstraction of raw water for drinking water production—this means that the metalimnetic phytoplankton populations need to be observed closely in order to adjust offtake depths and/or water treatment. An increasing understanding of the combined impacts of stratification stability, water transparency and nutrient concentrations may allow the development of models that can predict the wax or wane of this species in such waterbodies [146].

### 3.3.2. Benthic and Tychoplanktic Cyanobacteria

Benthic (toxic) cyanobacteria are often reported to have increased both in frequency of occurrence as well as distribution, though in part this perception might be due to increased awareness, triggered by dog poisoning that typically receives broad public attention [147]. In contrast to planktic cyanobacteria, for which mass developments are linked to higher trophic levels, benthic cyanobacteria are frequently reported from clear-water rivers and lakes [148] in which light can penetrate to the bottom, allowing mats of, e.g., *Phormidium* (*Microcoleus*) to grow or allowing macrophytes to develop which provide surfaces for epiphytic cyanobacteria. Nutrient mobilisation from co-occurring bacteria and fungi can enable benthic cyanobacteria to cover large areas of riverbeds [147].

The shift from planktic to benthic production in the course of water quality improvement and higher transparency has been shown for large European rivers such as the River Loire and the Danube [149,150]. Recent data from the River Loire indicate an increase of

*Phormidium* at some sites [151]. *Phormidium* (including synonymous taxa [152]) is a globally widespread largely benthic and often toxic cyanobacterium, reported from many rivers in, e.g., New Zealand [147], France [148] and California [153].

Potentially toxic benthic cyanobacteria such as *Phormidium*, *Microcoleus*, *Geitlerinema*, *Oscillatoria* or *Microseira wollei* are also reported from lakes and reservoirs in which water quality has improved from eutrophic to meso- or oligotrophic [154–156]. Lowe [157] identified the littoral of oligotrophic lakes with ample light as the preferred habitat of benthic cyanobacteria. As with benthic cyanobacteria in rivers, information on the abundance of benthic cyanobacteria half a century ago, prior to eutrophication, is scarce. Examples include two German waterbodies with a massively reduced P load since the 1980s which now have reached an oligo- to mesotrophic state, with planktonic cyanobacteria scarcely occurring: both lakes now show recurrent abundance of benthic cyanobacteria. In the Wahnbach reservoir (used for drinking water production) eutrophication control led to a drastic reduction of *P. rubescens*, but thereafter sporadic episodes with geosmin producing cyanobacteria inhabiting the littoral between 1 and 11 m occurred [158]. In Lake Tegel (Berlin), macrophytes are re-colonising the littoral as a result of increased water transparency, but in 2017 tychoplanktic *Tychonema* producing anatoxins was found in tufts of water moss (*Fontinalis* sp.) following an episode of dog poisoning [159] and since then is leading to recurrent dog death incidents.

Benthic cyanobacteria appear to have received attention particularly after such incidences, and thus the timepoint of their initial occurrence and of conditions favouring their abundance cannot be elucidated retrospectively. While optimal light conditions certainly are important, hydrophysical conditions such as storm-driven detachment of submerged macrophytes or water level fluctuations in reservoirs may further influence their abundance and in particular, whether or not they appear in parts of a water body where people or pets might be exposed.

## 4. How Do Trophic and Climatic Changes Affect Health Risks from Cyanotoxins?

Health risks from cyanobacterial blooms depend on the levels of biomass that they attain and on the contents of various toxin types in that biomass. Toxin contents in turn depend on the taxonomic composition of a bloom, as the potential to synthesise specific types of toxins is highly variable between cyanobacterial taxa (species and genera). Some toxin types such as microcystins are produced by a high variety of taxonomically distant taxa, while the number and diversity of taxa potentially producing cylindrospermopsins is much lower. While the different specific combinations of environmental conditions favouring the proliferation and dominance of particular taxa is well described [160,161], for inferring the likely toxin types and concentrations, these at best allow worst-case estimates based on the upper range of known toxin/biomass ratios. This is because the taxonomic category of a 'species' encompasses a range of different genotypes which can vary considerably in their content of secondary metabolites, including cyanotoxins [142,162–164].

Factors determining toxin contents—and hence toxin concentrations—discussed in this section act on different levels of regulation, schematically illustrated in Figure 2. An impact of nutrient availability on cellular toxin content is under discussion particularly for N and the peptide toxins, the microcystins, and in Section 4.1.1 we review the current state evidence for this, as well as approaches for clarifying this question. However, on the level of distinction between genotypes it is still poorly understood which environmental conditions favour which genotypes producing particular cyanotoxins, and in consequence impacts of changes in environmental conditions on dominance of genotypes with and without the production of certain toxins are still largely unclear. While we address this in Section 4.1.2, understanding why certain genotypes with specific metabolites dominate under certain conditions requires a broader approach, including the wide array of secondary metabolites in cyanobacteria that are synthesised by similar pathways and possibly have similar functions in producing strains. It is still enigmatic which competitive advantage some cyanobacterial taxa gain from the metabolically costly production of a huge variety of

secondary metabolites, and toxicity to vertebrates is rather a coincidence than a result of evolutionary selection—cyanobacteria produced microcystins long before vertebrates have evolved [165]. In Section 4.2 we discuss whether the key groups of cyanotoxins are now known with the four major groups for which the World Health Organisation provided further guideline values in 2020 [10]. Section 4.3 explains the considerations leading to their derivation, as these are important for assessing the implications that the in situ concentrations observed may have for human health. Section 4.4 discusses how the species shifts to metalimnetic and benthic habitats caused by oligotrophication change risks of human exposure to cyanotoxins.

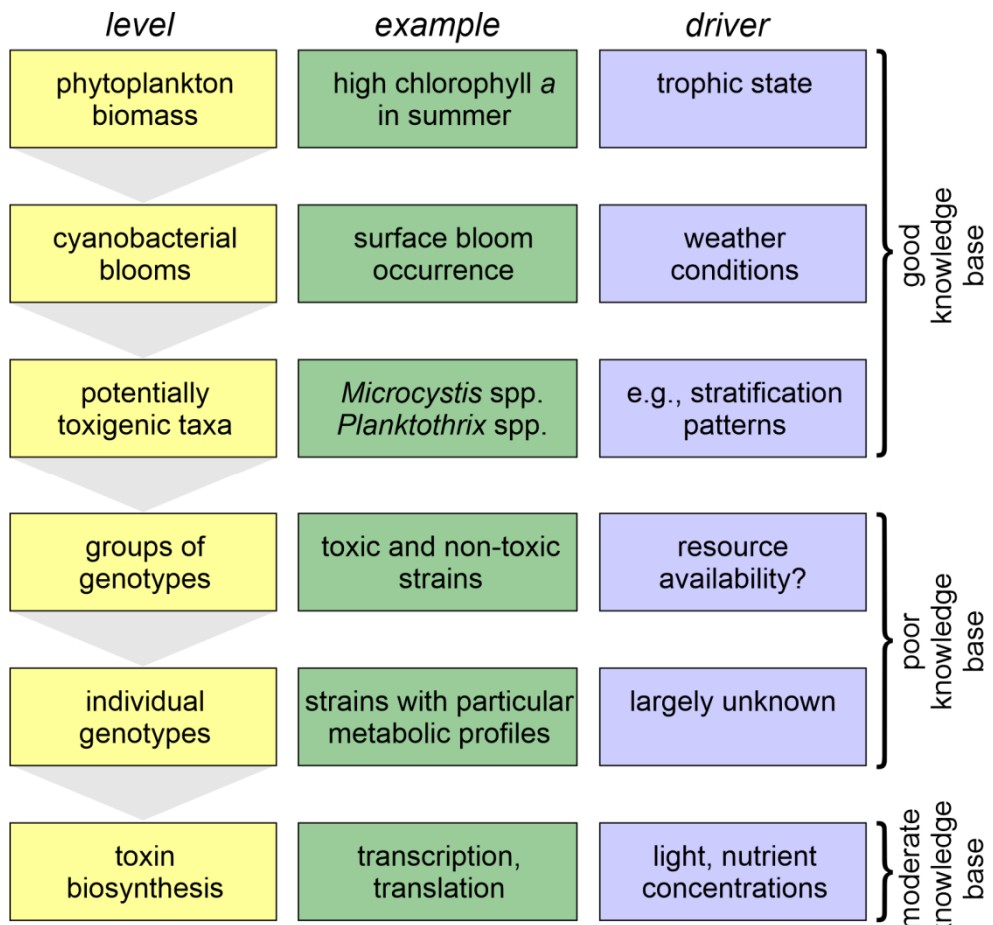

**Figure 2.** Levels at which environmental conditions drive cyanotoxin occurrence and concentrations.

### 4.1. Impact of Changes in the Availability of a Resource on Cyanotoxin Concentrations

For the first three levels of the environmental conditions driving cyanotoxin occurrence and concentrations illustrated in Figure 2, a good understanding is attained, providing a sound basis for developing management measures to control bloom occurrence. For the following two levels, the drivers of genotype occurrence, scientific knowledge is scarce, while for the lowest level, the intracellular processes of toxin biosynthesis are moderately well understood.

Differences in toxin production between individual genotypes adds further levels of complexity to the understanding of toxin occurrence in waterbodies. This limits the prediction of in situ toxin concentrations from biomass levels of the producing species to worst-case estimates. Many studies refrain from breaking taxonomic identification down below the genus level because of the difficulties of species distinction and identification (as well as ongoing changes in taxonomy [152]). However, for studying toxin occurrence yet a further differentiation below the species level is important because within most

potentially toxigenic species genotypes with and without genes for the production of specific metabolites coexist and co-occur in natural populations. For an understanding of the dynamics of toxin concentrations, a distinction only between toxigenic and non-toxigenic genotypes is presumably an oversimplification.

Toxigenic genotypes, this is, those that possess functional genes for the biosynthesis of a particular toxin, can be further sub-divided into clones or clonal complexes with a specific metabolome (i.e., producing individual structural variants of toxins and other peptide metabolites), and most likely also by specific ranges of cell quota (i.e., the amount of individual structural variants per cell).

A number of studies have assessed the diversity of genotypes within natural populations, showing that generally multiple toxigenic as well as non-toxigenic genotypes coexist that are often indistinguishable by conventional analysis such as light microscopy [23,166–168]. However, most of these studies reflect only snapshots of individual or a few populations while the dynamics of genotype occurrence are rarely described [169–173]. Still fewer studies have addressed the competition between genotypes and if so, in most cases with a focus on only one toxic and one non-toxic strain (Table 1). To our knowledge, no studies have been published on the competition of multiple toxic and non-toxic genotypes in an experimental system of a complexity that approaches that of natural systems. Any extrapolation of results obtained from an experimental setup of low complexity, for example, co-culture of one toxic and one non-toxic strain, risks being premature.

### 4.1.1. Cellular Level of Peptide Content and Biosynthesis

For the lowest level in Figure 2, the genetic background [174] and the cellular regulation of peptide biosynthesis, a wealth of studies is available ranging from conventional culture studies to more recent molecular approaches focusing on the transcriptional response, for example, to light intensity [175], iron availability [176] or nutrient stress [177]. The growth conditions that have been found to influence toxin biosynthesis vary between studies, and they do not show coherent outcomes: results vary between species, strains and experimental conditions, showing that stress through resource limitation or temperature can lead to more or to less microcystin produced per cell. Moreover, few experiments have been done with continuous or semi-continuous cultures, and only in these is limitation constant and clearly focused on one resource while in batch cultures, conditions (in particular, photon flux density) change as cell density increases.

The results of these many culture studies do, however, allow two overarching insights: one is that for microcystins and cylindrospermopsins the variability of toxin content in response to experimental conditions rarely exceeds a factor of 2–4 [142,162]. The other is that the majority of studies agree on the finding that no factor or growth condition has been found that completely suppresses toxin production (with few exceptions [162]), in line with the observation that microcystin production is coupled to growth rate [178]. This is remarkably different from regulation of comparable secondary metabolite production in filamentous fungi, where a large share of biosynthetical gene clusters is silent under certain conditions [179] and secondary metabolite production is subject to complex regulation, including epigenetics [180]. Therefore, toxin production by a cyanobacterial strain generally results in a toxin content within a comparatively narrow range that seems to be strain-specific. This means, for example, that one strain produces microcystins at cell quota of 5–20 fg/cell while in another strain of the same species the range is 50–200 fg/cell. Presence of biosynthesis gene clusters such as the *mcy*-cluster in a strain that has not been found to produce microcystins is, thus far, only confirmed for strains with a mutation in the gene cluster [126]. Future studies on various factors affecting toxin production by cyanobacteria most likely will not produce results that fundamentally overturn what has been found in hundreds of preceding studies.

The more recent field of ecological stoichiometry has addressed cellular microcystin content in the wider context of analysing the impact of nutrient availability on the relative atomic composition of biomass, emphasising that nitrogen amounts to 14% of the molecular

mass of microcystins. It thus seems plausible that N-limitation should reduce the cellular content of microcystins, and indeed a number of laboratory culture studies report this finding (e.g., [22,181]; see also the discussion in Gobler et al. [182]). However, others find the opposite, e.g., Peng et al. [183] report an inverse relationship of average microcystin content per cell to N concentration from experiments at 20–30 °C but not at lower temperatures.

Brandenburg et al. [184] conducted a meta-analysis for N-rich toxins in a range of freshwater and marine genera from 37 published laboratory culture studies of cellular toxin contents in response to N and P limitation. The authors conclude that "*responses of N-rich toxins to nutrient limitation were less consistent than previously reported*": cellular toxin contents decreased by only about 60% under N limitation, and this was statistically significant only when pooling the data for all 37 studies across all toxins and all freshwater and marine phyla. For the 24 studies on microcystins alone, extremes ranged from almost 3-fold decrease to almost 3-fold increase, and while some decrease was visible in the data for about 11 of these studies, this was not statistically significant. Additionally, three of these studies showed an increase and about 10 showed no response of cellular microcystin contents to changes in N-availability. Brandenburg et al. [184] also evaluated 16 studies addressing the impact of P-limitation on cellular microcystin content, and here they found more pronounced responses, with an overall increase by 88% and only two studies showing a decrease. A further important result of this meta-analysis is that cellular content of N-rich toxins increased or decreased in parallel with the overall N-content of biomass, this is, with C:N ratios and N:P ratios. This finding is interpreted as reflection of N being primarily allocated to population growth and not specifically to the production of N-rich metabolites.

This finding highlights that a focus on the metabolites that happen to be toxic to vertebrates is too narrow an approach for the target of elucidating the environmental conditions driving cellular cyanotoxin production and content. This pertains in particular to oligopeptides: several hundreds of other oligopeptides from cyanobacteria have been described [12], all with a relative N-content similar to that of microcystins, many with some sort of bioactivity at least demonstrated in vitro [185]. Broadening the perspective beyond microcystins to cover the entity of N-rich secondary metabolites suggests that the availability of dissolved inorganic N (nitrate, urea or ammonia) is unlikely to generally favour microcystin production over that of other cyanopeptides. Rather, this broader perspective requires analysing changes in the total content of the cells' oligopeptides and possible shifts in their relative proportions. While the high in situ diversity of cyanobacterial peptides has been acknowledged for nearly two decades [186–189], very few studies have quantified oligopeptides other than microcystins [11] because this broader approach is hampered by the lack of standards for quantifying other cyanopeptides. Natumi and Janssen [190] addressed this gap for multiple peptide classes, including microcystins, in *Microcystis* PCC 7806 and *Dolichospermum* NIVA-CYA 269/6 by recording relative changes and thus showed that the cellular peptide content responded synchronously to changes in nutrient concentrations.

### 4.1.2. Genotype Level of Peptide Diversity and Dominance

As reviewed in Suominen et al. [191], the differences in microcystin content between genotypes by far outweighs differences induced by resource limitation or temperature. Co-occurrence of genotypes with high and low or no cellular contents of specific metabolites in field populations explains why the microcystin cell quota found in field samples are typically several-fold lower than the maximum cell quota observed in some strains in the laboratory [142,192]. As indicated in Figure 2, at the level of individual genotypes within cyanobacterial populations, knowledge about drivers of dominance is scarce. A number of publications report the sub-specific in situ diversity of bloom forming taxa such as *Planktothrix*, *Microcystis* and *Aphanizomenon* either in individual or multiple waterbodies [23,127,193–196] by various typing schemes. Further, genotype diversity has been studied on larger geographic scales [127,197]. Based on the available studies it seems fairly safe to assume a multitude of co-existing genotypes in any cyanobacterial bloom—or to

consider a bloom formed by a single or only a few genotypes an exception. Only a few studies have assessed the seasonal dynamics of multiple genotypes within cyanobacterial populations [169,170,173,198] or shifts over the course of years [23,172]. Although respective studies in this field would be exciting, methodological problems need to be solved to allow a cost-effective, high-resolution monitoring.

To elucidate drivers of genotype composition and dominance, approaches that are well established at the species level in phytoplankton ecology can be applied on the genotype level, using molecular tools to distinguish a few genotypes within a species. For the species level Litchman and Klausmeier [199] review how traits affect the outcome of competition, depending on conditions in the waterbody; i.e., resistance to grazing and parasites, modes of reproduction and survival of inocula, temperature-dependence of growth rates and growth responses to resource limitation. Among these traits, studies of differences between cyanobacterial genotypes have chiefly addressed growth rate responses to temperature and resources (for resistance of cyanobacteria to grazing, see Moustaka-Gouni and Sommer [53]).

Data characterising uptake affinity and cellular storage have become available from a large number of experiments, compiled, for example by Litchman et al. [33] and Edwards et al. [77] who used such data to model the outcome of competition between species. At the level of different cyanobacterial genotypes such data are rare. Differentiation at this level may be more challenging because cell size and shape are regarded to be the "master variable" that governs resource uptake [200], but the size and shape of strains within a species of cyanobacteria usually show no difference. Indeed, Hesse and Kohl [201] found no difference in maximum growth rates relative to light of a wildtype and a microcystin-deficient mutant of *Microcystis* PCC 7806 grown in semi-continuous cultures, and Briand et al. [202] found no difference between the maximum growth rates ($\mu_{max}$) of five strains of *Planktothrix agardhii* (three of which produce microcystins). However, some studies addressing the species level show that not all resource affinity parameters necessarily correlate with cell size and shape [77,199]; niche separation of the morphologically very similar *Planktothrix rubescens* (red, metalimnetic) and *P. agardhii* (green, epilimnetic) may serve as an example. Additionally, monoculture experiments with different strains of cyanobacterial species have found distinct differences in their growth traits: Suominen et al. [191] report differences in growth characteristics between the *Microcystis* PPC 7806 wildtype and *mcy*-knockout mutant, and these differences were sufficiently distinct to model the outcome of competition between both strains with these parameters.

Differences in nutrient or light acquisition characteristics between genotypes may be too subtle to show as statistically significant in any of the growth parameters listed above, but nonetheless determine the outcome of competition. In such cases, co-culture experiments can be a more effective approach to elucidate their relative competitive power under specific combinations of growth conditions, either by taking field populations into micro- or mesocosm cultures to test which genotypes end up dominating, depending on how conditions are manipulated, or by combining two culture strains to see who 'wins'. Results of respective studies are compiled in Table 1.

**Table 1.** Competition between strains: examples of outcomes of competition experiments between cyanobacterial strains with (MC+) and without (MC−) microcystin production or pairs of wildtype (*mcy*+) and knockout mutants (*mcy*−). The studies are organised by the limiting resource or growth condition, respectively (some studies occur twice).

| Strains; Culture Method | Limitation and Growth Condition | Dominance Attained in Competition Experiments | Comments | Reference |
|---|---|---|---|---|
| *M. aeruginosa* **MC+**: V163; NIVA CYA140 **MC−**: V145; NIVA CYA43; chemostat cultures | Light 25 µE/m$^2$ × s | **MC−** for both strains within <2 weeks for the 2 strains with pronounced difference in critical light intensity; within >160 days for the 2 strains with similar critical light intensity | Strains differentiated semiquantitatively using DGGE bands, different pigments as well as microcystin concentrations | [203] |
| *Microcystis* sp. **MC+**: UTCC 300; **MC−**: UTCC 632; UTCC 633; batch cultures | Light: 20 versus 80 µE/m$^2$×s | **MC+** at low light **MC−** at high light in co-culture with UTCC 632, but **±balance** at high light in co-culture with UTCC 633 | In monoculture at low light no difference in growth rate between strains, but at high light higher growth rate for MC+ | [204] |
| *M. aeruginosa* PCC 806: **MC+**: wildtype **MC−**: *mcy*− mutant; batch cultures | Light: 39 and 5 µE/m$^2$ × s | *mcy*− at optimal light; **±balance** at low light | In monoculture growth characteristics of both strains were not significantly different; allelopathy of *mcy*+ strain tested and not observed | [181] |
| *Microcystis* sp. **MC+**: FACHB905 **MC−**: FACHB469; batch cultures | Light: 35–120 µE/m$^2$ × s Temperature: 16–32 °C | **MC+** under all light and temperature conditions tested | In monoculture MC− grew faster than MC+; independently of growth condition an un-identified allelochemical of FACHB905 (not MCs!) seemed to suppress FACHB469 | [22] |
| *Microcystis* isolated from Kinneret **MC−** (brown) **MC+** (green); batch cultures | Temperature: 14.5–25 °C | **MC+** at lower temperatures; **MC−** at higher temperatures | In line with field observations from Lake Kinneret: in recent years with higher temperatures, brown (phycoerythrin-containing) non MC− producing *Microcystis* dominate | [23] |
| | Salinity: 150–1000 mg/L | **±balance** up to 11 days | | |
| *Planktothrix agardhii* **MC+**: 3 strains **MC−**: 2 strains; batch cultures | Light and temperature | **MC+** at low light and low temperature; **either MC−** or **balance** of both strains at high light and temperature | Interpretation as trade-off between better fitness of MC+ under limiting conditions and metabolic costs of MC− production under non-limiting conditions; Result in line with field observations of genotype composition during population growth | [202] |
| | N | **MC+** under N limitation; **MC−** after N pulse | | |

**Table 1.** *Cont.*

| Strains; Culture Method | Limitation and Growth Condition | Dominance Attained in Competition Experiments | Comments | Reference |
|---|---|---|---|---|
| *M. aeruginosa* PCC 7806 **MC+**: wildtype **MC−**: *mcy−* mutant; batch cultures | N | *mcy−* at optimal light (39 μE/m² × s) and low N; ±**balance** of both strains at low light, low N | In monoculture growth characteristics of both strains were not significantly different; Allelopathy of *mcy+* strain tested and not observed | [181] |
| *M. aeruginosa* PCC 7806 **MC+**: wildtype **MC−**: *mcy−* mutant; chemostat and semi-continuous cultures | N | *mcy+* always wins: if limitation is continuous, N is applied as single strong pulse or at 3-day intervals | Adding P pulsewise leads to co-existence of wildtype and mutant | [191] |
| *Microcystis* sp. **MC+**: FACHB905 **MC−**: FACHB469; batch cultures | N, P | **MC+** or **balance** at lower N or P | In monocultures **MC−** grew faster than **MC+**; Unidentified allelochemical of FACHB905 seems to suppress FACHB469 under all growth conditions tested | [22] |

The only striking observation gleaned from the compilation in Table 1 is the contradictory nature of "who wins", including counter-intuitive outcomes such as the dominance of the non-microcystin-producing strain under N-imitation in the study by Suominen et al. [202]. The results of LeBlanc Renaud et al. [204] show that under the same co-culture conditions the microcystin producer can outcompete one non-producing strain but not the other.

Basic data on resource acquisition characteristics and competitive power are best obtained under steady-state conditions, preferably in continuous or semi-continuous cultures. Beyond this basic characterisation it is also important to test pulse-wise or fluctuating changes in resource availability because this is likely to enable co-existence, as has been shown on the level of competition between species [109,205]. In nature, pulse-wise changes in resource availability are the rule and the most likely explanation for coexistence. Briand et al. [202] tested the response of co-cultures of strains of *P. agardhii* first grown in N-deplete medium to which a pulse of nitrate was then added, and this led to a switch in dominance from the microcystin-producing strain to the non-producer. In a similar experiment with a wildtype of *Microcystis* and a *mcy*-knockout mutant, Briand et al. [181] found a single pulse of nitrate to strengthen the pre-existing dominance of the non-producing mutant. With the same wildtype and mutant strains, but using semi-continuous cultures instead of batch cultures with nutrient pulses at 3-day intervals, Suominen et al. [191] found the opposite response to pulses of nitrate, i.e., dominance of the producing wildtype, but coexistence for up to 30 days in response to pulses of phosphate.

The outcome of competition between strains with and without a specific metabolite may or may not be related to a biological function of that metabolite—it may well be due to other differences between the strains. For example, Lei et al. [22] propose that the reduced competitive strength of a microcystin-producing strain (at high light intensity, low concentrations of N and P and low temperature) may have been caused by reduced production of an (unidentified) allelochemical sequestered by the non-producing strain (Table 1); they discuss this relative to other publications, some of which found evidence for such effects and some of which did not.

For the wide array of cyanopeptides in cyanobacteria it is indeed conceivable that the various peptides either have a similar or a complementary function. Briand et al. [206] discuss this as one possible interpretation of their results with a *Microcystis* wildtype producing microcystins and its knockout mutant: the latter contained significantly higher

concentrations—up to 12-fold—of a range of other oligopeptides, although the only difference between both strains was the lack of a functioning gene for microcystin production. These authors also found significantly higher cellular concentrations of oligopeptides when co-culturing strains than in monocultures. Interestingly, pure microcystin-LR was not the substance triggering these responses. Suominen et al. [191], referring to the experiments by Briand et al. [202], with *P. agardhii* suggest that growth conditions may not determine the outcome of competition directly, but possibly indirectly, through affecting the production of (as of yet unidentified) allelopathic substances.

We summarise the current state of knowledge on drivers of genotype composition as follows:

- The numerous monoculture experiments subjecting strains to different resource limitations or temperatures have not revealed any specific conditions that unambiguously enhance or reduce cyanotoxin production across the range of strains and species thus tested. Even nitrogen limitation seems to have little impact on cyanopeptide production beyond its general impact on the nitrogen content of cellular biomass.

- In field populations strains producing microcystins generally seem to co-exist with those that do not, and this observation is supported by the outcome of co-culture experiments run with pulsed nutrient application: pulses promote co-existence rather than dominance of one strain. A hypothesis worth testing is that this applies to other cyanobacterial oligopeptides as well.

- While all cyanotoxin groups include several congeners and the microcystins include many, bloom-forming cyanobacteria often also contain numerous further secondary metabolites, particularly peptides, of unknown function. Research focusing merely on the production of microcystins is likely to fall short of recognising potential trade-offs, complementary or substituting functions for the producing genotypes.

- The "chemical cross-talk" of peptides indicated by the results of some co-culture experiments opens exciting perspectives for understanding the biological function of these metabolites. This applies equally to the possibility that peptide metabolites confer resistance to fungal and possibly other parasites [207,208].

In conclusion, co-culture experiments seem to be the currently most promising approach for developing a better understanding of the function of cyanobacterial metabolites. However, when limited to two strains, a producer and a non-producer of a single type of metabolite, the outcome may well be coincidental, and with another combination of strains results may well be different.

After decades of research that so far has not found an unambiguous answer to the question of the biological function of cyanotoxins, cyanobacterial oligopeptides and other secondary metabolites, contributions to understanding the drivers of genotype composition are likely to be of substantial scientific value in their own right. Whether or not such insights are useful for planning management strategies then remains to be seen

### 4.2. Can We Expect New Cyanobacterial Toxins in the Future?

The number of studies and publications about toxic cyanobacteria has been steadily increasing over the past few decades. Similarly, although to a lesser extent, the search for natural products with pharmacological potential has led to the characterisation of a large number of cyanobacterial metabolites, some 2000 of which are listed in CyanoMetDB [12]. Despite these two developments, the number of cyanobacterial toxins has been stable over the years, in particular regarding major toxin groups rather than individual toxin variants. These groups, microcystins, cylindrospermopsins, anatoxins and saxitoxins, make up the largest share of known cyanobacterial toxins. Besides these, a few individual toxins have been described, such as guanitoxin or lyngbyatoxins. For microcystin variants alone, about 280 structural variants have been characterised [209] and new variants are still being found. One anatoxin-a variant, dihydro-anatoxin-a has been recently recognised to be abundant in many samples of benthic cyanobacteria [210], often exceeding the abundance of anatoxin-a.

A tentative conclusion from several decades of cyanotoxin research may be that chances for discovering a new group of cyanotoxins which occurs widely in aquatic (or terrestrial) ecosystems are low, and that toxic cyanobacterial compounds yet to be discovered are likely to be confined to cyanobacterial taxa that themselves are not widely distributed or that are produced by only a limited number of genotypes of more widely distributed taxa.

An example of a recently discovered toxin with (yet) regionally limited occurrence is aetokthonotoxin, a polybrominated tryptophane derivative that is suspected to be the cause of the death of bald eagles and water birds in the United States [211]. Brominated metabolites are known primarily from marine cyanobacteria [212] while halogenated metabolites in freshwater cyanobacteria are generally chlorinated [213]. Aetokthonotoxin is produced by *Aetokthonos hydrillicola*, an epiphytic cyanobacterium that morphologically resembles *Fischerella* sp. (with true branching) and that is found associated with the invasive water plant *Hydrilla verticillata* which is spreading throughout the southeast of the United States [214]. Many bioactive metabolites have been reported from the group of cyanobacteria to which *Aetokthonos* belongs (classically termed "Stigonematales", including, for example *Fischerella*, *Stigonema*; [215]), but since most genera of this group do not form water blooms, so far only few studies describe toxin production [216]. At present, aetokthonotoxin appears to be confined to a comparatively low number of aquatic ecosystems and it remains to be seen whether it will reach a broader distribution.

Another cyanobacterial compound that earned considerable attention as a possible potent toxin during the last decade is β-methylaminoalanine (BMAA), a non-proteinogenic amino acid. However, a number of inconsistencies in respective studies, in particular with respect to accuracy of detection and quantification [217,218], led Chernoff et al. [219] to the conclusion that, based on available data, BMAA does not pose a relevant health risk for human populations. This conclusion does not deny the toxic effects observed in a number of bioassays, where BMAA primarily causes neurological damages. The main critique of Chernoff and co-workers is that doses applied in bioassays were orders of magnitude higher than what could be assumed as realistic exposure. This point of view has been challenged by Dunlop et al. [220], and it remains to be seen whether clear evidence can be produced that BMAA can lead to adverse health effects at exposure levels consistent with its occurrence in water environments (and quantified with appropriate techniques).

This highlights a general problem: the uncertainty what should be considered a toxin. In toxicology, a toxin is not only defined by the observation of toxic effects in bioassays, rather, the effects need to be observable when testing doses that are not too far above realistically possible exposures [221]. In other words, many compounds are indeed toxic at sufficiently high dose, and only if exposure pathways exist that can cause a toxic dose under realistic exposure scenarios—including uncertainty factors considered in the calculation of guideline values (see below)—is a compound legitimately classified as toxin. Many compounds essential for healthy human nutrition show adverse or toxic effects at doses substantially exceeding the usual uptake: for example, doses of some essential micronutrients such as vitamin A [222] or zinc [223] only 10 times the recommended daily uptake may already result in adverse health effects. Labelling a cyanobacterial compound as 'toxin' requires assessment by experienced toxicologists, with toxicity data preferably evaluated by a team. Further, distinction is relevant between toxicity to mammals, including humans, and to aquatic animals such as planktonic Crustacea.

A further aspect is that in vitro effects such as enzyme inhibition, toxicity to isolated cells or to membranes do not allow the conclusion of toxicity to an entire animal, in which excretion and detoxification mechanisms may well prevent the substance from reaching the organs or cells that showed sensitivity in vitro. For this reason, derivation of guideline values or standards is still based on whole animal studies performed following standardised OECD protocols (reviewed in Lawton et al. [221]).

In contrast, there seems to be a recent trend to "upgrade" known cyanobacterial metabolites from "bioactive" to "toxic". For example, Sieber et al. [224] describe a "toxic

chymoptrypsin inhibitor" that showed toxicity in *Thamnocephalus* bioassays "roughly one order of magnitude weaker than other cyanobacterial toxins". Although it is of substantial interest to study cyanobacterial metabolites, many of which were isolated because they show bioactivity [225], a labelling as toxin should be done with diffidence. Similarly, Roy-Lachapelle et al. [226] wrote "*Harmful algal blooms of cyanobacterial origin have the potential to generate hundreds of secondary metabolites referred to as cyanotoxins*", thus suggesting that any cyanobacterial compound is a toxin—which is simply not true. Of the hundreds of secondary metabolites found in a cyanobacterial bloom, only a small share are true toxins, although most compounds or compound classes may have been shown to be bioactive in various assays [227,228]. It is important to limit the classification of cyanobacterial metabolites as 'toxins' to those for which exposure can indeed potentially be harmful in order to focus public health capacity on those hazards which indeed can cause a risk to human health.

### 4.3. Which Cyanotoxin Concentrations Actually Present a Threat to Human Health?

The provisional guideline value of the World Health Organisation (WHO) for Microcystin-LR in drinking water of 1 µg/L has been widely used to discuss the health implications of microcystin concentrations found in waterbodies, not only by limnologists but also by public health professionals. When concentrations of up to 2.5 µg/L were found in the drinking water of Toledo, USA, in August 2014, for two days half a million consumers were advised not to drink the water [229]. Optimising drinking water treatment reduced concentrations below 1 µg/L, and the treatment plant was later upgraded to better cope with cyanobacterial blooms. However, besides the costs of supplying bottled water to this population, consequences of the two days of 'do not drink' advisory, such as closure of restaurants and schools, has a major impact on people's lives and income, raising the general question whether health risks from brief exposure to concentrations in the lower 1-digit range justify such consequences and whether immediately focusing all available funding on upgrading treatment would not be the better, more sustainable option. The incident thus spotlighted the need for short-term guideline values. Additionally, the more recent availability of toxicological data on cylindrospermopsin now allowed derivation of guideline values for this toxin. In response, the WHO expert group on chemicals in drinking water re-assessed the available information and developed further cyanotoxin guideline values, including for short-term exposure via drinking water and recreation. These now cover all four major cyanotoxin groups—microcystins MCs), cylindrospermopsins (CYNs), Anatoxin-a and congeners (ATXs) and saxitoxins (STXs). WHO also gives background documents explaining the considerations for their derivation [230–233]. In Section 4.3.1 we briefly summarise the background of these derivations and in Section 4.3.2 we discuss their implications for threats to human health.

#### 4.3.1. Current WHO Guideline Values for Four Groups of Cyanotoxins and How They Are Derived

Chemicals in drinking water are generally regulated very conservatively. For those with a toxicity threshold and derived from long-term animal experiments, a factor of 1000 is frequently used between the dose that caused no effects (NOAEL—No Observed Adverse Effect Level) and the tolerable daily intake (TDI). This is based on a factor of 10 for each of three possibilities: (i) higher individual sensitivity (intraspecies variability), (ii) higher sensitivity of humans as compared to the laboratory animals (interspecies variance) and (iii) uncertainties in the toxicological data base. For (iii) the factor can be lower if the toxicological data are comprehensive or if these are supported by epidemiological data (as is the case for STX). These three factors are multiplied to ensure sufficient protection even if all three conditions apply simultaneously. However, the probability of all three of these possibilities to apply to a given substance is low, and should one possibility actually require a higher factor, this is likely to be balanced by one or both of the other two needing only a lower factor. Thus, multiplying all three factors ($10 \times 10 \times 10 = 1000$) gives a high margin of safety. For further information on guideline derivation, see WHO [234,235].

Figure 3 illustrates the derivation of the provisional WHO guideline value for microcystin-LR in drinking water from a 13-week mouse study [236]: no treatment-related effects were observed at a dose of 40 µg/kg per day (NOAEL), while at the highest dose tested (1000 µg/kg per day) hepatic lesions typical for MC-LR appeared in all animals, and at the lowest observed adverse effect level (LOAEL) of 200 µg/kg per day some animals showed slight hepatic damage.

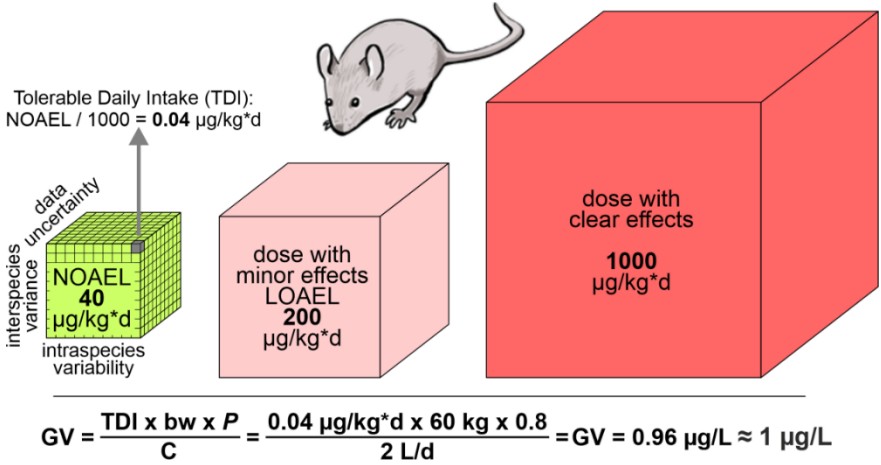

**Figure 3.** Schematic illustration of the derivation of tolerable daily intakes (TDI) and guideline values (GV) for lifetime daily exposure to 2 L of drinking water. The values for LOAEL and NOAEL are from the 13-week mouse study for Microcystin-LR by Fawell et al. [236]. The leftmost cube illustrates the derivation of TDI from NOAEL by applying uncertainty factors for interspecies variability ($\times$10), intraspecies variance ($\times$10), and data uncertainty for less than lifetime exposure ($\times$10). From TDI the GV is derived by assuming a bodyweight of 60 kg, a proportion of uptake via drinking water of $P = 0.8$, and a daily water consumption of $C = 2$ L/day.

Table 2 shows the current WHO guideline values together with the assumptions WHO used to derive them, that is, the NOAEL, bodyweights, uncertainty factors, the volumes of water to which people are likely to be exposed as well as the proportion of exposure allocated to different pathways. Note that for MC and CYN these guideline values are provisional due to limitations of the toxicological data, including data being available only for one key congener (probably the most toxic one) for each toxin group. For short-term and recreational exposure, extrapolation from the NOAEL uses lower uncertainty factors for gaps in the toxicological data because uncertainties when extrapolating from part of a lifetime in the animal experiment to a full lifetime do not apply for an exposure of up to two weeks. For Anatoxin-a and its variants (ATXs), in an experiment designed similarly to that for MC-LR quoted above, none of the dose levels tested indicated damage due to ATX-a, and thus no NOAEL could be derived. However, the data do allow giving an upper bounding level below which toxic effects are very unlikely, termed 'health-based reference value'. For saxitoxins (STXs), data for chronic exposure are lacking, but for acute exposure they are available from incidents of food poisoning with marine shellfish. Note that the acute guideline value for STXs in drinking water is derived on the basis of the bodyweight of an infant and all recreational values are based on the bodyweight of a small child. Refer to the WHO background documents [230–233] for more information on the considerations leading to these derivations as well as guidance on their application, including the understanding of 'short-term' exposure. Note also that while the derivation is based on toxicological data for one main congener (due to lacking data for the others), WHO recommends application to the sum of all congeners present in a sample.

**Table 2.** WHO cyanotoxin guideline values and health-based reference values, respectively, and assumptions used for their derivation. MCs: microcystins; CYNs: cylindrospermopsins; ATXs: anatoxins; STXs: saxitoxins.

| Toxin and Exposure Pathway | NOAEL (or LOAEL) | Uncertainty Factor | Partitioning Relative to Other Exposure Pathways | Water Volume Potentially Ingested | Bodyweight | Guideline Value (Provisional for MC and CYN) |
|---|---|---|---|---|---|---|
| MCs, drinking water lifetime | 40 µg/kg × day | 1000 | 80% | 2 L | 60 kg | 1 µg/L |
| MCs, drinking water short-term * | 40 µg/kg × day | 100 | 100% | 2 L | 60 kg | 12 µg/L |
| MCs, recreation | 40 µg/kg × day | 100 | 100% | 0.25 L | 15 kg | 24 µg/L |
| CYNs, drinking water lifetime | 30 µg/kg × day | 1000 | 80% | 2 L | 60 kg | 0.7 µg/L |
| CYNs, drinking water short-term * | 30 µg/kg × day | 300 | 100% | 2 L | 60 kg | 3 µg/L |
| CYNs, recreation | 30 µg/kg × day | 300 | 100% | 0.25 L | 15 kg | 6 µg/L |
| ATXs, drinking water short-term * | 98 µg/kg × day | 100 | 100% | 2 L | 60 kg | 30 µg/L ** |
| ATXs, recreation | 98 µg/kg × day | 100 | 100% | 0,25 L | 15 kg | 60 µg/L ** |
| STXs, drinking water acute | 1.5 µg/kg × day (LOAEL) | 3 | 100% | 0.75 L | 5 kg | 3 µg/L |
| STXs, recreation | 1.5 µg/kg × day (LOAEL) | 3 | 100% | 0.25 L | 15 kg | 30 µg/L |

* Short-term exposure refers to about two weeks, is not intended for multiple periods per season and is linked to informing the public, recommending bottled water for infants and small children as well as initiation of measures to remediate the situation. ** Health-based reference value as upper limit.

### 4.3.2. Cyanotoxins in the Context of Other Health Hazards in Water

Among the hazards occurring in water, the public health impact of pathogens generally far outweighs that of chemicals: for 2016, WHO [237] estimates almost two million deaths worldwide from infectious diseases attributed to quantifiable effects of inadequate water, sanitation and hygiene, amounting to 3.3% of all deaths, and in terms of disease burden to 4.6% of all disability-adjusted life years (DALYs). In contrast, for chemicals in water the WHO Guidelines for Drinking-water Quality [234] discuss that generally these rarely occur in concentrations causing adverse health effects unless exposure occurs for prolonged periods of time, that is, in the range of years. Additionally, a scan through the WHO background documents for many chemicals in drinking water shows that the chief exposure pathways to most of the substances are air and food. For recreational water quality the WHO guidelines note that reports of human health impacts associated with chemicals in water are rare [238].

Acute illness attributable to exposure to chemicals in water is largely limited to incidents of massive accidental contamination and even in these, there may be little exposure through ingestion because such water is likely to be "*undrinkable owing to unacceptable taste, odour and appearance*" [234]. While bloom-containing water is indeed not palatable, drinking water treatment may improve palatability by removing much of the organic matter, and if treatment is not appropriately designed and managed, efficacy of cyanotoxin removal may be limited, leading to exposure [239]. Recreational exposure may lead to substantial cyanotoxin ingestion, and for this exposure pathway, cyanotoxins may well be the most relevant group of chemicals [238]. Criteria for assessing short-term exposure through drinking water and through recreation are therefore important.

Interestingly, chronic exposure to chemicals in drinking water at levels causing demonstrable illness is chiefly known from natural geogenic sources, in particular for arsenic and fluoride, in some regions also uranium [234]. While cyanotoxins are natural substances, in most cases their occurrence in health-relevant concentrations is not natural, but due to widespread anthropogenic eutrophication, and depending on climatic and hydrophysical

conditions, blooms may last for many months [192]. Among the chemicals in water to which people are exposed, cyanotoxins may therefore be those occurring most widely.

The overall evidence for illness attributed to chemicals in water also applies to cyanotoxins: although some are very potent and massive bloom situations can lead to high concentrations, there are not many cases of well-substantiated evidence for cyanotoxins as cause of human intoxication (reviewed in Humpage et al. [240], for drinking water and by Chorus and Testai [241] for recreational exposure). Where symptoms are unspecific and not typical for the respective cyanotoxin(s), such as nausea, diarrhoea or skin irritation, other agents may as well have been the cause, including pathogens, possibly associated with the bloom or reaching the water together with nutrients feeding the bloom. Prematurely dismissing these may risk not finding the real cause, and WHO discusses criteria for attributing health complaints to cyanotoxins (see the introduction to chapter 5 in Chorus and Welker [10] and WHO [238]).

In consequence, a low one-digit number of μg/L of cyanotoxins does not imply a 'serious threat' to human health if it occurs only for a few days, particularly if such concentrations are found in a waterbody and not in drinking water. However, blooms cause a high load of organic material, and where treatment does not effectively remove this from drinking water which then is chlorinated, they can cause elevated concentrations of chlorination by-products, some of which are suspect carcinogens and cause unpleasant taste and odour, thus causing further health concerns.

### 4.4. Implications of Shifts to Benthic or Metalimnetic Cyanbacteria for Human Exposure

As discussed in Section 3.3, increased water transparency through oligotrophication opens habitats in deeper water layers and on the sediment surface, and these may be populated by benthic, periphytic or metalimnetic cyanobacteria. Dog deaths from ingestion of beached material at recreational sites or water in a drinking water reservoir deeply brownish-red from *Plankothrix rubescens* draw substantial public attention and cause concern. Such phenomena challenge water managers, particularly if they appear as water gets clearer in consequence of investments in nutrient reduction. However, for risks of human exposure the substantial differences in spatial distribution within the waterbody are relevant: epilimnetic blooms can cause high cyanotoxin concentrations particularly if they form scums, and where scums accumulate in bays used for recreational activities, particularly bathing, toxin concentrations can reach a range of mg/L, risking acute symptoms through oral uptake [241]. In contrast, *Planktothrix rubescens* resides in the metalimnion during the summer, and total waterbody mixing occurs outside the main season for recreational waterbody use, rendering recreational exposure risks less likely.

For lumps of detached floating or beached benthic material also, risks of oral uptake appear unlikely. Dog deaths occur because—in contrast to humans—these animals are attracted to this material and thus ingest substantial amounts. However, concentrations in the surrounding water are typically low [159] because anatoxin-a released by lysing cells is quickly diluted well below concentrations in the range of the WHO health-based reference values (30 μg/L for drinking water and 60 μg/L for recreational water; see Table 2). Dilution is particularly relevant in streams in which water flowing over benthic mats will remove released ATXs, but also in lakes and reservoirs where water volumes are large relative to the amount of such material. Thus, while scenarios of concern cannot totally be excluded (e.g., children playing with lumps of beached material or persons ingesting this in near-drowning accident situations), human health risks from recreational exposure to benthic cyanobacteria appear much lower than those from surface scums. Based on the same considerations, this also applies to drinking water production. Wood [242] developed recommendations for informing the public in New Zealand to avoid contact and particularly oral uptake of mats or lumps of benthic cyanobacteria with particular emphasis on small children as vulnerable group.

In contrast, risks of cyanotoxins from *P. rubescens* in drinking water offtakes depend on the specific situation: where thermally stratified reservoirs are the drinking water source

and drinking water offtakes are constructed with variable depths, the risk of abstracting water from the metalimnetic layers of this species can be minimised. These layers are typically quite thin and distinct and can be avoided by adjusting the offtake depth [243]. Where adequate systems of monitoring and management responses are in place, cyanotoxin risks from *P. rubescens* can thus be more readily controlled than those from epilimnetic blooms: removing the latter requires elaborate treatment not only addressing toxins, but typically also the high concentration of organic substances caused by them [239].

In consequence, where (re-)oligotrophication of waterbodies shifts species to metalimnetic or benthic cyanobacteria, cyanotoxin health risks are likely to be lower than those from epilimnetic blooms during the hypertrophic or eutrophic phase of the respective waterbody.

## 5. Summary and Conclusions

The state of knowledge that we reviewed here is necessarily incomplete, largely omitting influences of catchment conditions as well as of food-webs, and only touching upon the complex mechanisms driving nutrient release from sediments. We do, however, believe that it sufficiently highlights the need for careful differentiation of the messages we derive regarding the three questions we ask in the introduction. For these we propose the following conclusions:

1.  Climate change, in particular warming, is likely to promote blooms in some waterbodies but not in all. In waterbodies which lack the carrying capacity to sustain high phytoplankton biomass, elevated water temperatures may chiefly affect species composition and dynamics rather than total biomass. Effects of climate change may be more pronounced in shallow than in stratified waterbodies because in the former indirect effects such as sediment resuspension during storms are stronger. In deep stratified lakes, climate change can work both ways—rendering conditions for cyanobacterial proliferation more favourable or less favourable. Therefore, statements generally claiming that climate change will '*globally*' lead to '*ever increasing*' blooms are neither scientifically sound nor helpful for management.

2.  Controlling cyanobacterial blooms can definitely be successful without also reducing N, provided the concentrations of P can be brought down to sufficiently low levels. The outcomes of the last century's research still hold: if one nutrient limits the carrying capacity for biomass, excess concentrations of the other cannot promote significant further growth. Not the N:P ratio, but the absolute concentration of the limiting nutrient determines the maximum possible biomass of phytoplankton and thus of cyanobacteria. It follows that whether N or P are currently limiting biomass can be inferred from their absolute concentrations.

    Focusing investment on measures to reduce one nutrient is likely to be more effective than spreading the available funding across measures addressing both. Whether the reduction of P or N is more efficient is specific to the waterbody, its catchment, and socioeconomic conditions.

    We emphasise that there is no question about the need to reduce the overall emissions of N into the environment in face of the dramatic increase of emissions in many regions of the world, not only into waterbodies but also into terrestrial ecosystems and groundwater. Strategic planning needs to include assessments of the wider environmental impacts of local remediation measures.

3.  Trophic and climatic changes will affect health risks from cyanobacteria. Trivially, conditions leading to more cyanobacterial biomass can increase risks of exposure to toxins. Health risks depend on the amount and time span by which cyanotoxin concentrations exceed WHO guideline values for the respective exposure pathway and time span. Whether trophic and climatic changes will increase health risks or not depends on local conditions.

It is well understood that the cellular up- or downregulating of the production of toxic metabolites is of minor relevance as compared to the genotype composition of field populations. However, we are only beginning to understand the drivers of genotype dynamics. Understanding these requires experiments comparing their growth rates under tightly defined conditions (preferably in (semi-)continuous cultures), including competition experiments between strains producing different metabolites. This implies combining approaches and techniques developed in the 20th century with the new, 21st century methods.

The current evidence for the impact of environmental conditions on toxic genotype occurrence is too contradictory and the understanding gleaned from the data too poor to draw conclusions for management. Premature conclusions for management are neither helpful nor necessary. Rather, the focus on controlling the occurrence of cyanobacteria as such is preferable because the sheer biomass of blooms is detrimental to aquatic ecosystems and compromises human use of waterbodies.

**Author Contributions:** Conceptualisation, I.C., J.F. and M.W.; writing—original draft preparation, I.C., J.F. and M.W.; writing—review and editing, I.C., J.F. and M.W.; visualization, I.C. and M.W. All authors have read and agreed to the published version of the manuscript.

**Funding:** This research received no external funding.

**Institutional Review Board Statement:** Not applicable.

**Informed Consent Statement:** Not applicable.

**Data Availability Statement:** Not applicable.

**Acknowledgments:** Special thanks are due to Max Tilzer for training in the IBP-inspired fundamentals of production biology in the 1980s and very recently, to Hans Paerl and Mark McCarthy for fruitful controversial discussion of the concept of limitation. The thorough review and helpful suggestions to improve the manuscript by four anonymous reviewers is greatly appreciated.

**Conflicts of Interest:** No conflict of interest.

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
