# Peer review of "Cyanobacteria and Cyanotoxins in a Changing Environment: Concepts, Controversies, Challenges"

_water, doi:10.3390/w13182463_

Round 1

Reviewer 1 Report

The paper “Cyanobacteria and Cyanotoxins in a Changing Environment: Concepts, Controversies, Challenges” authored by Chorus et al., reviewed the effect of climate change and nutrients to the increasing Cyanobacteria bloom and Cyanotoxins, as well as further influence to the health risks of exposure. This work is well organized and will benefit the readers and researcher in in related study fields. However, the difference between the effect of climate change and nutrients to algal bloom and Cyanobacteria bloom should be separated, particularly in the further discussion about the Cyanotoxins and climate change and nutrients. Some specific issues of Cyanobacteria bloom or the relation between algal bloom and Cyanobacteria bloom should be externalized. Some Specific comments:

1) the effect of climate change and nutrients to Cyanobacteria bloom and Cyanotoxins expressed by Mate analysis could be much more clear.

2) the effect of climate change to Cyanobacteria bloom and Cyanotoxins with the cross trophic state is confused with the section 3, some descriptions could be a little bit more specific.

3) line 552 and line 876, both of these figures are labeled by figure 1, this should be revised and the corresponding contents also should be revised.

4) line 225 Chl.-a should be Chl-a.

5) line 353 “2.”should be “3.”line792 “3.”should be “4.”line 882 4.1.1.. should be 4.1.1., line 656, “:” should be removed. other similar blemishes should be revised.

6) how the relationship or different effect of total nutrient and dissolved nutrient to the growth Cyanobacteria. Some internal relations of influencing factors should be clear and could be introduced.

7) The format of references should be unified. So many blemishes in the references.

Author Response

please see file attached

Reviewer 2 Report

This study review a terrific amount of publications to discuss three important issues. How cyanobacterial blooms respond to the climate change in different types of waterbodies? Whether double control of N and P are necessary for mitigating cyanobacterial blooms? How trophic and climate changes affect health risks caused by toxic cyanobacteria. I have no doubt about the huge efforts that the authors have made, and totally agree that those three issues are very important. In general, the MS is well written and easily understood. I just have a few suggestions for further improvement.

Introduction: The title of this MS is “Cyanobacteria and Cyanotoxins in a changing environment…”. Thus, I would recommend to reorganize this part, since the logic thread was not clear enough. Firstly, describe the changing trend of the key environmental factors, e.g. temperature, N and P concentration, ect. Then, state how cyanobacterial composition and cyanobacterial bloom would change. Finally, introduce the story about the toxic cyanobacteria and toxin.

Line 184-188: Actually, large shallow eutrophic lakes and some reservoirs are very sensitive to the heavy rain caused by extreme weather. For example, the frequency of the extreme weather events-induced extended blooms significantly increased after 2012 in Lake Taihu, since strong winds and heavy rainfall resulted in a short-term increase in nutrient concentrations (Yang et al., 2016 Water Research). I thinks it should cite more published work.

Line 353: The sequential number should be 3. Meanwhile, I am wondering the subtitle doesn’t match the title of MS, which is not relevant to the control or mitigating cyanobacterial bloom. I understood that you should elaborate how N and P concentration in waterbody change currently and in future. The changing pattern might be affected by the treatment process of sewage water, and also the strategy to control non-point pollution. Then, under this background, how cyanobacterial bloom will change.

Line 792: The sequential number should be 4.

Author Response

please see file attached

Reviewer 3 Report

Dear Editor,

This manuscript is a Review on Cyanobacteria and Cyanotoxins with a special focus on some controversial thoughts.  As a review, the data presented in the manuscript are not novel, however, the authors show flaws in studies and also point out that the community should refrain from overall generalizations since each body of water is unique.

However, the authors are very selective in the studies they show, which could be seen by some as 'cherry picking' data.  During revision, I'd ask the authors to take that into account. Also, I feel the authors should add more information on CO2 and climate change, with the example of 2 papers below:

Visser et al. (2016) How rising CO 2 and global warming may stimulate harmful cyanobacterial blooms Harmful Algae 54: 145-159. doi: 10.1016/j.hal.2015.12.006.

Van de Waal (2011) Reversal in competitive dominance of a toxic versus non-toxic cyanobacterium in response to rising CO2. The ISME Journal 5, 1438–1450

In general I find this manuscript important to add to the current body of knowledge on cyanobacteria and recommend publication after Revision.  

Comments:

Verify the way references are cited as they are cited more than one way in text.

Make sure all Latin words are in italics.

There are 2 figure 1's.

In Section 4.2, please also cite Breinlinger et al. 2021 Science 371 (6536): eaax9050 DOI: 10.1126/science.aax9050, since this explains aetokthonotoxins.

Line 38: Delete repeated word: than

Line 41: Change 'will occur increasingly' to 'will increase'

Lines 41 and 43: Use of 'in consequence' is repetitive.  Consider using 'due to' in line 41.

Line 54: Are you speaking of just between two species are among several species? Revise usage if necessary.

Line 62: Anatoxin-a(S) has been renamed guanitoxin, please update. (see Fiore et al 2020 om Harmful Algae)

Lines 67-69: I don't fully understand what the authors are conveying in this sentence; it is a run-on and confusing. Please break it up into more than one sentence.

Lines 67-74: Refs?

Lines: 75-79: REFs?

Lines 83-84: REFs?

Lines 102-104: I think the list should be 1, 2, 3 and not 4, 5, 6.

Lines 102 and 108: I'd use the plural 'waterbodies' instead of the singular 'waterbody'.

Line 114: ecological what?  I think there is a word missing.

Line 125: add a close parenthesis ) after 2005.  

Line 125: COles and JOnes in text is cited 2005, but in REFs is 2000. 

Line 126: change 'curves of growth rate' to 'growth rate curves'

Line 126: There is no Reynolds 1997 in references.

Line 127 and 128: 'are' should be 'is' since the reference is single author, which also means 'they' should become 'he' in the sentence.

Line 130: Add eukaryotic to '6 eukaryotic algal species'

Line 136: Please update to Raphidiopsis and Dolichospermum, which are the correct names of the genera.

Line 146: Delete comma (,) after Lei

Lines 146-148: This sentence is a little confusing.  Microcystins were the cause of what?

Line 178: add 'a' to .. as A potential ...

Line 196: Please correct this typo: 'can not only can'

Line 201: Change 'that' to 'than'.

Line 205: 'neural' not 'neuronal'

Line 207: delete 'thus'

Line 206-210: This is a run-on sentence and is hard to understand what it is saying.  Please revise it and maybe break into 2.

Line 219: delete 'for' before shallow waterbodies.

Line 234: Anabaena should be Dolichospermum if the authors are referring to the planktonic bloom former.

Line 237: delete: indeed

Lines 271-276: Run-on sentence and a bit confusing.  Maybe break it up into 2?

Line 290: Independent not Independently 

Line 293: year for Posch et al.?

Line 313: ... where stratification is not Schauser ... : what do you mean by not Schauser?  I think there is a word missing.

Line 316: delete: it

Line 394: replace: 'to consider' with 'to be considered'

Line 431: delete: in

Line 460: add an 'n' to 'norther' -> should be 'northern'.

Line 470: verify usage of () around author's name

Lines 473 & 475: change ':' to '.'

Figure 1: This figure can be a bit confusing since the color/no color is not explained.

Line 641: add: 'of' -> ... source of internal

Line 773: N-limitation not N-limiation <- spelling

Lines 834-837: I don't understand this sentence. It seems a bit verbous, especially here: "that is, down to the drivers of species’ occurrence and biomass, a good understanding is attained to a degree that allows a translation of the available wealth of knowledge"

Line 846: What do the authors mean by ambiguous taxonomy?

Line 876: This should be Figure 2, not 1, right?

Line 883: Figure 2 or 1? Also, add 'toxin' to 'cellular regulation of TOXIN biosynthesis'.

Line 923: add: 'of' after 'range'

Line 971: I think you are referring to Figure 2

Line 1037: delete: 'in' after the word 'show'

Line 1168: anatoxin-a(s) is now guanitoxin, please update (Fiore et al. 2020)

Lines 1350-1355: In subtropical and tropical areas, blooms can last all year long especially due to poor sewage treatment, large urban areas, and high agricultural production in many of these areas.  As half of the world's population live in these areas, it might be important to comment on this.

Lines 1384: I disagree that benthic proliferations are easier to control. We are in the infancy of understanding the controlling factors of benthic cyanobacteria, how to manage them (including efficacious treatments), and the diversity of compounds they produce.

Line 1394: verify use of esp., maybe write it out?

Line 1472: Genus is Geitlerinema not Geilerinema.

Line 1473-1474: Maybe list the genera in alphabetical order?

Line 1475: verify spelling of 'oligogropxhi'

line 1506: 'this' not 'thi'

Line 1543: 'hypereutrophic' not 'hypertrophic'

Line 1622: 'is of minor' not 'is if minor'

Line 2039: Vidal et al is a separate reference so should be on new line.

Author Response

please see file attached

Reviewer 4 Report

General comments:

This is a long, and mostly comprehensive summary of some of the issues with cyanobacteria and cyanotoxins. I found the section on the large number of peptides produced by cyanobacteria in the form of secondary metabolites to be especially informative and important to the audience of this article.

The material is well-presented for the most part. A few of the sections were unclear, and those are noted in the specific comments below. By the Summary and Conclusions of this long piece, the authors must have been very tired, as the writing becomes more confused.

The authors state that “concern is increasing that climate change may compromise chances for such success”. The summary should come back to this point in a more informative way. There is little synthesis concerning the role that climate and climate change plays in cyanobacterial growth.

The punctuation throughout the manuscript needs to be revised. Check for the use of colon and semicolon, which are frequently used incorrectly. In most places a period to divide phrases into two separate sentences is appropriate.

Specific comments:

Line 30. Correct to “epilimnion”

Line 57. Omit “already”, replace “their” with “toxin”

Line 75. Suggest revising to “We now have a general, albeit incomplete, picture of the specific toxins produced by particular cyanobacterial taxa in particular environmental conditions by geographic region.”

Line 101. Correct numbering from “4,5,6” to “1,2,3”

Line 125. Insert missing “)”

Line 146. Correct to “Lei et al. (2015)”

Line 165. Replace “:” with “.”

Line 185. I think “precede” is incorrect, do the authors mean “follow”?

Line 213. The citation for Reichwaldt & Ghadouani needs the year included

Line 284. A comma is needed after “Garda”,

Line 319. Incorrect use of colons!!

Line 341. Incorrect use of colon!!

Line 353. I don’t understand the scheme for numbering headers. Something seems to be off.

Line 368. What is meant by “difficult to handle”? Be specific.

Line 386. Incorrect use of semicolon. Replace with “.”

Line 407. Remove colon

Line 408. The statement that phytoplankton, in general, are able to quickly take up nutrients needs clarification. State the taxonomic groups that the citation refers to.

Line 414. Correct punctuation for citation

Line 445. Replace “scatter” with “variation in response”, or another phrase that is descriptive

Line 473. Colon

Figure 1. The caption needs more explanation. Include that the green circles represent algal cells (or biomass) and their component ratio (7:1) of N:P.

Line 616. Check correct citation of Lewis

Line 633. This paragraph on internal and external sources of P is crucial. However, I read it several times, and I still fail to understand the author’s point. I fail to follow why mineralization is “faster under oxic conditions”. Explain. What is an example of “internal measures to intercept P release”?

Lines 645-648. These sentences are too awkward and vague to understand. Clarify.

Line 664. Edit to “1980’s”

Line 801. Citations need years.

Line 954 – and other places. Check the form of citations, as many need to be corrected for the position of the parentheses.

Line 1251. Define PAC

Line 1270. Unclear what is meant by “i.e. strictly”.

Line 1330. Not worth providing acronym (DALY) if it is not used again, or used much further away in text that a reader would forget its meaning.

Line 1337. Why is acute underlined?

Line 1362. Correct spelling of “diarrhoae”

Line 1368. Correct “a low one-digit numer of μg/L of cyanotoxins” and reword to simplify.

Line 1375. Uneven use of italics in headers.

Line 1386. A prominent example of what? Unclear.

Line 1394. Unclear reference of “nutrient loads changed their”. Sounds like nutrient loads are changing something, when you mean deep lakes have changed in trophic state. This paragraph is difficult to follow, I suggest rewriting to begin with stating why this is an important/different/worthwhile case. As written, the paragraph is a sort of smattering of statements with no clear thesis.

Line 1415 and elsewhere. Make sure liter is “L”, rather than “l”. Species names need to be consistently in italics.

Line 1464. Correct punctuation in citations.

Line 1473. This paragraph does not seem to be completely written – incomplete references, misspellings, etc.

Line 1505. It is unclear why P. rubescens is given such mention, but not other species.

Line 1508. State the concentrations, rather than “two-decimal range”.

Line 1510. State that the “oral uptake” is by dogs, or rewrite.

Line 1541. Uncomprehensible paragraph.

Line 1555. It seems odd to begin these conclusions with “yes”, as there is no reference to a question being asked.

Summary and Conclusions. This section needs to be rewritten to be clear and concise. It seems the authors were tired after writing this long article.

Author Response

please see file attached
